# Photochemical box-modelling of volcanic $SO_2$ oxidation: isotopic constraints

Tommaso Galeazzo[1,2], Slimane Bekki[1], Erwan Martin[2], Joël Savarino[3], and Stephen R. Arnold[4]

[1]LATMOS/IPSL, Sorbonne Université, UVSQ, Université Paris-Saclay, CNRS, Paris, France
[2]ISTeP, Sorbonne Université, CNRS, Paris, France
[3]IGE, Univ. Grenoble Alpes, CNRS, IRD, INP-G, 38000 Grenoble, France
[4]Institute for Climate and Atmospheric Science, School of Earth and Environment, University of Leeds, Leeds, UK

*Correspondence to:* galeazzo.tommaso@latmos.ipsl.fr; tommaso.galeazzo@gmail.com

**Abstract.** The photochemical box-model CiTTyCAT is used to analyse the absence of oxygen mass-independent anomalies (O-MIF) in volcanic sulphates produced in the troposphere. An aqueous sulphur oxidation module is implemented in the model and coupled to an oxygen isotopic scheme describing the transfer of O-MIF during the oxidation of $SO_2$ by OH in the gas-phase, and by $H_2O_2$, $O_3$ and $O_2$ catalysed by TMI in the liquid phase. Multiple model simulations are performed in order to explore the relative importance of the various oxidation pathways for a range of plausible conditions in volcanic plumes. Note that the chemical conditions prevailing in dense volcanic plumes are radically different from those prevailing in the surrounding background air. The first salient finding is that, according to model calculations, OH is expected to carry a very significant O-MIF in sulphur-rich volcanic plumes and, hence, that the volcanic sulphate produced in the gas phase would have a very significant positive isotopic enrichment. The second finding is that, although $H_2O_2$ is a major oxidant of $SO_2$ throughout the troposphere, it is very rapidly consumed in sulphur-rich volcanic plumes. As a result, $H_2O_2$ is found to be a minor oxidant for volcanic $SO_2$. According to the simulations, oxidation of $SO_2$ by $O_3$ is negligible because volcanic aqueous phases are too acidic. The model predictions of minor or negligible sulphur oxidation by $H_2O_2$ and $O_3$, two oxidants carrying large O-MIF, are consistent with the absence of O-MIF seen in most isotopic measurements of volcanic tropospheric sulphate. The third finding is that oxidation by $O_2$/TMI in volcanic plumes could be very substantial and, in some cases, dominant, notably because the rates of $SO_2$ oxidation by OH, $H_2O_2$, and $O_3$ are vastly reduced in a volcanic plume compared to the background air. Only cases where sulphur oxidation by $O_2$/TMI is very dominant can explain the isotopic composition of volcanic tropospheric sulphate.

## 1 Introduction

Volcanic activity is one of the major natural forcers of the Earth's climate, as volcanic emissions alter the chemical composition and radiative properties of the atmosphere, at local, regional and even global scales (Stocker et al., 2013; Langmann, 2014). Beyond their environmental impacts, sulphuric acid aerosols have adverse effects on human health since they are linked to cardiovascular and respiratory diseases (Pope III, 2002; World Health Organization, 2009). Moreover, sulphate aerosols can lead to acid rain causing damage to vegetation and to urban infrastructures. Over the last decades, our understanding of volcanic

emissions in the atmosphere have greatly improved, thanks to satellite and field measurements, and to more sophisticated physical-chemical models (Robock, 2000; Bobrowski et al., 2003; Mather et al., 2003; Textor et al., 2004; Roberts et al., 2009; von Glasow, 2010; Roberts et al., 2012, 2014). The main gases emitted to the atmosphere by volcanic activity are respectively $H_2O$, $CO_2$, $SO_2$, $H_2S$, and halogen species, such as HCl, HBr and HF (Textor et al., 2004; Rose et al., 2006; Oppenheimer et al., 2013). In addition, measurements at crater rims of volcanoes suggest also direct emissions of small amounts of sulphate aerosols (Allen et al., 2002; De Moor et al., 2013).

Among all the compounds emitted, volcanic sulphur gases, and in particular $SO_2$, are considered to be the most effective in affecting climate. Climatic perturbations from volcanic emissions are principally caused by conversion of sulphur gases into sulphate aerosols, which can then interact with solar and terrestrial radiation via scattering and absorption (Stocker et al., 2013). Once injected into the troposphere, volcanic $SO_2$ is converted in few days typically to $H_2SO_4$ by a range of gas-phase and liquid-phase reactions taking place in volcanic plumes and clouds (Chin and Jacob, 1996; Stevenson et al., 2003a). In the atmosphere, depending on the oxidation pathway, $H_2SO_4$ is produced either in the gas phase or liquid phase. When generated in the gas-phase, volcanic $H_2SO_4$ condenses very rapidly onto pre-existing particles, or it may even form very small sulphate particles by nucleation. In the boundary layer, sulphate aerosols have a residence time much shorter than a week because of the fast wet and dry depositions. However, at higher altitudes, such as in the free troposphere, removal is much slower; consequently, volcanic sulphate aerosols can have a much longer residence time of up to a few weeks (Stevenson et al., 2003b, a; Kristiansen et al., 2016). In addition, the residence time of volcanic aerosols in the stratosphere can reach lifetimes of about a year (Thomason, L. and Peter, 2006).

Nowadays, anthropogenic $SO_2$ emissions outweigh those from natural sources (Smith et al., 2011). Volcanic quiescent degassing and eruptions is an important natural source of $SO_2$, notably to the free troposphere (Bates et al., 1992; Graf et al., 1998). Volcanic emissions release about 10-13 $Tg \cdot y^{-1}$ of $SO_2$ to the atmosphere (Andres and Kasgnoc, 1998) and contribute to up to 10% to total sulphur emissions to the atmosphere (Stevenson et al., 2003a). Remarkably, volcanic emissions also have a bigger impact on the tropospheric aerosol burden than other sulphur sources (Graf et al., 1998) because volcanoes tend to emit $SO_2$ at higher altitudes than most other surface sulphur emissions, where the lifetime is longer.

Most of the tropospheric sulphate is generated in the liquid phase (Alexander et al., 2009) via oxidation of aqueous $SO_2$ by dissolved oxidants of the atmosphere, such as $H_2O_2$, $O_3$, $O_2$ catalysed by transition metal ions ($Fe(III)$ and $Mn(II)$) and, possibly HOBr and HOCl (Vogt et al., 1996; von Glasow et al., 2002; Stevenson et al., 2003a; Berglen et al., 2004; Park et al., 2004; Alexander et al., 2009; von Glasow and Crutzen, 2013; Chen et al., 2016). Note that the importance of the halogen oxidation pathway remains unclear. A significant amount of tropospheric $H_2SO_4$ is formed in the gas phase via the termolecular reaction between $SO_2$ and hydroxyl radicals (OH) (Calvert et al., 1978). In presence of liquid water and for typical pH values of atmospheric water droplets ( $3.0 < pH < 5.6$), $SO_2$ is quickly oxidized by dissolved $H_2O_2$, and the two species rarely coexist in liquid phases (Gervat et al., 1988; Chandler et al., 1988; Daum et al., 1990; Zuo and Hoigne, 1993; Laj et al., 1997). At acidic pH values, synergism among transition metal ions (TMI) enhances the rate of $SO_2$ oxidation by dissolved $O_2$ (Brandt et al., 1994; Brandt and van Eldik, 1995), which can thus compete with the other $SO_2$ oxidation channels. Particular attention has been paid recently to this heterogeneous oxidation pathway, since its contribution could have been underestimated in previous

budget assessments of sulphate production in the troposphere (Alexander et al., 2009; Goto et al., 2011; Harris et al., 2013). During eruptive events, volcanoes emit large quantities of ash and coarse material rich in iron-minerals (mainly glass, and in lesser extents magnetite and hematite), which can easily dissolve in water because of the high acidity reached in volcanic cloud droplets and aerosols (Ayris and Delmelle, 2012; Hoshyaripour et al., 2015; Maters et al., 2016). As a consequence, the $O_2$/TMI heterogeneous oxidation reaction may be more significant than previously thought.

Quantifying the importance of the different $SO_2$ oxidation pathways is challenging. It requires the quantification of, among other things, the rates of the different oxidation processes. Conventional methods rely mostly on models that are evaluated and constrained with atmospheric concentration measurements of oxidants, because there is no direct means of measuring chemical fluxes associated with individual reactions (Morin et al., 2008). Simultaneous measurements of $SO_2$ oxidants in both the gas- and liquid phases in the atmosphere, let alone specifically in a volcanic plume, would be experimentally challenging. Alternative approaches need to be considered to reduce the uncertainty in the relative contributions from the different oxidation pathways. Isotopic approaches can provide such constraints (Brenninkmeijer et al., 2003; Thiemens, 2006). Isotopic ratios, indeed, provide direct insights into the nature and importance of individual oxidation fluxes (Savarino et al., 2007; Morin et al., 2008; Martin et al., 2014).

Thanks to peculiar distribution of isotopes among its three oxygen atoms, ozone and its chemistry provides a useful tool of investigation for atmospheric processes using isotopic signatures. Ozone bears a very significant non-mass dependent (also called mass-independent) isotopic fractionation, which is due to its formation mechanism (Heidenreich III et al., 1986; Krankowsky et al., 1995; Marcus, 2013). Since oxygen atoms in tropospheric oxygen-bearing species sometimes originate directly or indirectly from ozone via multiple photochemical reactions, a variety of atmospheric species carry anomalous isotopic mass-independent fractionations (MIFs) (Thiemens, 2006). For oxygen-bearing species, the anomalous oxygen MIF ($\Delta^{17}O$, O-MIF) is calculated with respect to a reference standard:

$$\Delta^{17}O = \delta^{17}O - 0.52 \times \delta^{18}O \tag{1}$$

Where $\delta^{17}O$ and $\delta^{18}O$ represent deviations to the reference standard isotopic ratios ($R_{std}$):

$$R_x = \frac{^xO}{^{16}O} \qquad x = 17; 18 \tag{2}$$

And:

$$\delta^xO = \frac{R_x}{R_{std}} - 1 \tag{3}$$

Ozone is a key chemical reactive species of the troposphere. Its isotopic anomaly is intrinsically generated (through photolysis and recombination reactions) instead of being inherited by isotopes transfer like for most atmospheric species (Marcus, 2013). Other oxygen-bearing species in the atmosphere can gain excess-$^{17}O$ by transfer of this ozone anomaly via reactions with ozone itself, reactions with species that have already inherited the ozone anomaly or via anomalous kinetic isotopic effect (Röckmann, 1998; Lyons, 2001; Michalski et al., 2003). As a consequence, transfer of oxygen MIF among atmospheric species is process-specific and can be used as a signature to trace the chemistry of species as they react with specific oxidants. Once

the isotopic anomalies of the oxidants are characterised, the resulting $\Delta^{17}$O of an end-oxidation product is simply a linear combination of the isotopic signatures of all the oxidation channels weighted by their respective contributions, to the total production of the end-oxidation products. During the last decade, there has been an increasing number of studies that have used O-MIF oxygen anomalies in oxidation products to constrain oxidation channels, often coupling isotopic measurements

and photochemical isotopic modelling (Michalski et al., 2003; Alexander et al., 2005; Morin et al., 2008; Gromov et al., 2010; Michalski and Xu, 2010).

The isotopic signature in sulphates generated in the troposphere, the so-called secondary sulphate (by opposition to sulphate directly emitted in the atmosphere, the so-called primary sulphate) reflects the competition within different oxidation channels. In the liquid phase, sulphate oxygen MIF is produced during sulphur oxidation by transfer of isotopic anomalies from ozone

and $H_2O_2$, whereas sulphate with O-MIF very close to 0 ‰ is produced in the liquid phase via $O_2$/TMI oxidation (i.e. -0.08 ‰). Mass-dependent (MIF anomaly = 0 ‰) sulphate is generally produced via OH oxidation in the gas-phase (Savarino and Thiemens, 1999a, b; Savarino et al., 2000; Martin et al., 2014).

Most present-day tropospheric sulphates have O-MIF anomalies ($\Delta^{17}$O) of the order of 1 ‰ typically (Lee et al., 2001; Lee and Thiemens, 2001). However, there is some variability. For instance, O-MIF of sulphate aerosols generated in marine envi-

15 ronments are higher compared to isotopic anomalies found in continental sulphates (Alexander et al., 2005). Very significant $\Delta^{17}$O have also been found in volcanic sulphates collected from ash deposits dating back to the Miocene and the Oligocene, whose values reach 3.5 - 5.8 ‰. These peculiar isotopic anomalies have been linked to a different oxidative state of the atmosphere at that time (Bao et al., 2000, 2003). Tropospheric volcanic sulphates of the present era distinguish themselves from other tropospheric sulphates by having a $\Delta^{17}$O often close to 0 (within the measurement error of about 0.1‰). This feature is

20 found all over the world in sulphates collected from volcanic ashes of small and medium-size tropospheric explosive eruptions, independently from location, or geology of ash-deposits (Bao et al., 2003; Bindeman et al., 2007; Martin et al., 2014) (see Table:1). Notably, this is often the case for volcanic sulphate extracted from ash-deposits which are found very far from volcanoes, where secondary sulphate is expected to dominate. The only exception is volcanic sulphates in ice cores originating from very large volcanic eruptions. This sulphate had formed and transited through the stratosphere (Savarino et al., 2003; Baroni

et al., 2007).

The question is why tropospheric volcanic sulphate from volcanic ash-deposits does not appear to carry some isotopic O-MIF as for other types of tropospheric sulphates. One might expect that part of sulphate produced by tropospheric oxidation of volcanic $SO_2$ to carry some MIF isotopic anomaly because the dominant $SO_2$ oxidants in the troposphere are thought to be species carrying O-MIFs ($O_3$ and $H_2O_2$) with some contribution from $O_2$/TMI (Martin et al., 2014). An important difference

between volcanic sulphur and most other sources of sulphur is that it is often emitted within dense volcanic plumes whose chemical compositions are radically different from the background air. The purpose of the present box-modelling study is to explore in detail the oxidation and fate of volcanic sulphur in dense volcanic clouds and the resulting isotopic MIF signature in volcanic sulphate. The objective is to see to what extent the chemical environment of dense volcanic plumes may affect sulphur dynamics and pathways of oxidation and, hence, sulphate isotopic composition.The focus here is on volcanic clouds

that are rich in sulphur but poor in halogens, such in the case of intra-plate and rifting plate volcanoes (e.g. Nyarogongo in

Congo, Erta'ale in Ethiopia, Kīlauea in Hawai'i) (Aiuppa, 2009; Oppenheimer et al., 2013).Volcanic eruptions with remarkable low halogens to sulphur emissions are the Holuhraun (Bárðarbunga) eruption of 2012-2014 in Iceland (Ilyinskaya et al., 2017; Stefánsson et al., 2017), and the Kīlauea eruption of 2008 in Hawai'i (Mather et al., 2012). In particular, $HCl/SO_2$ ratios of the order of $10^{-2}$ have been observed for the Kīlauea eruption of 2008 (i.e. $HCl \approx$ 10-50 ppbv).

The second section of this work describes the photochemical model, including its sulphur heterogeneous chemistry scheme and the associated oxygen isotopic scheme. The mass balance equations used to evaluate the transfer of MIF oxygen anomaly from ozone to volcanic sulphate via different oxidation pathways are also presented. The third section is devoted to the study of individual and combined oxidation pathways and the resulting isotopic signatures in numerical experiments for this work standard volcanic plume conditions. The fourth section covers sensitivity model studies, investigating how different parameters

in volcanic plumes affect the final isotopic anomaly in sulphate. Dominant oxidation pathways are identified and the ability of the model to reproduce observed isotopic signatures of volcanic sulphate is assessed.

## 2   Modelling approach

The photochemical box-model used during simulation is the Cambridge Tropospheric Trajectory model of Chemistry and Transport (CiTTyCAT), a photochemical box-model developed to simulate tropospheric chemistry (Evans et al., 2000; Sander

et al., 2006; Real et al., 2007; Pugh et al., 2012). It describes the standard gas-phase photochemistry accounting for: kinetics of tropospheric species (bimolecular, termolecular, and photodissociation reactions), and deposition of gases and particles. Photolysis reaction rates are evaluated using the Fast-J code (Wild et al., 2000). Kinetic data are taken from JPL's datasheets (Sander et al., 2006). CiTTyCAT had already been used with success to constrain seasonal pathways of reactive nitrogen species in the troposphere, through the implementation of its chemical scheme with an isotopic transfer scheme accounting for $\Delta^{17}O$

production in nitrates (Morin et al., 2008). We have extended the capabilities of the model by including parameterisations of the transfer of soluble species between liquid and gas phases, of $SO_2$ heterogeneous chemistry, of pH in the liquid phase and of MIF of oxygen atoms in sulphates.

### 2.1   General continuity equations

The model resolves differential coupled mass balance equations (continuity equations) describing the time evolution of species

concentrations in the troposphere. For given initial (e.g. initial concentrations of species) and environmental conditions (e.g. pressure, temperature), mass balance equations are solved for each species, accounting for production and loss as follows:

$$\frac{dC_i}{dt} = P_i - L_i \tag{4}$$

where $C_i$ is the concentration of species $i$, $P_i$ the sum of physical and chemical production rates for $i$, and $L_i$ the sum of the

physical and chemical loss rates of $i$.

Production and loss terms are calculated using chemical reaction kinetics, where time evolution of concentrations of chemical species depends on the relevant rate constants ($k_i$) and on concentrations of reactants. They also include liquid-gas transfer and deposition. In addition, mixing of air between the volcanic sulphur cloud and the outside background air is also accounted for. It is parametrised by a simple linear relaxation scheme resulting in an exponential decay of plume concentrations towards background concentrations (Methven et al., 2006; Arnold et al., 2007).

$$\left(\frac{dC_i}{dt}\right)_{mixing} = K_{mix} \cdot (C_i - C_{i(bck)}) \tag{5}$$

where $K_{mix}$ is a first-order mixing rate coefficient representing all the processes mixing volcanic air with the background atmosphere and $C_{(i,bck)}$ is the concentration of species $i$ in the background air. $K_{mix}$ is set to 0.1 $day^{-1}$, a value typical of low mixing in the free troposphere and corresponding to a characteristic mixing timescale of 10 days (Methven et al., 2006; Arnold et al., 2007).

## 2.2 Liquid-gas mass transfer

Concentrations of relevant soluble species are calculated taking into account its partition between the gas and liquid phases. The transfer in both directions (evaporation, condensation) is dynamically computed. At each time step, rates of transfer are defined as:

$$\frac{d[C_{(aq)}]}{dt} = J_i \cdot \left(C_{(i)} - C_{(i,s)}\right) \tag{6}$$

where $C_{(i)}$ is gaseous concentration of species i far from liquid droplets, $C_{(i,s)}$ is the gaseous concentration of species i at the surface of droplets (which is assumed to be the equilibrium saturation vapour of $i$ over the liquid), and $J_i$ is the coefficient of condensation (from gas phase to liquid droplet) for species $i$, which is calculated using the Dahneke's expression (Dahneke, 1983) to cover mass-transfer from the continuum to the kinetic regime (see pg. 502 of Seinfeld and Pandis, 2016).

Throughout all the model simulations, droplets are assumed to be very large, with a radius of 5.0 µm. The sensitivity of the results to the assumed amount of liquid phase is explored varying the concentration of water droplets (and hence the liquid water content) instead of varying the size of droplets. It is also possible that emitted water condenses onto ash particles. Our treatment does not discriminate between liquid droplets and liquid phases at the surface of solid particles.

## 2.3 Gaseous and heterogeneous sulphur chemistry

The model already describes the $SO_2$ gas-phase chemistry. Since $SO_2$ is a mildly soluble species undergoing acid-base equilibrium in the liquid phase, we have added the gas-liquid transfer and the chemical reactions and equilibrium associated with its presence in the liquid phase (see Table:2). The extent of $SO_2$ dissolution into water droplets is controlled by the pH. The oxidation of S(IV) species ($HSO_3^-$, $SO_3^{2-}$, $SO_{2(aq.)}$) by reactions with $H_2O_2$, $O_3$ or $O_2$ in the liquid phase pushes the gas-liquid partition towards dissolution of gaseous $SO_2$. A diagram of the sulphur chemical model is presented in Fig.:1. Since the model CiTTyCAT resolves continuity equations for species with gas-phase concentration units, liquid phase concentrations (e.g. M) and rate constants have to be expressed into gas-phase units in the code in order to be treated by the CiTTyCAT chemistry solver (Seinfeld and Pandis, 2016).

The species involved in the acid-base equilibriums of $SO_2$ and $H_2SO_4$ are often grouped together according to their oxidation state:

$$S(IV) = SO_{2(g.)} + SO_{2(aq.)} + HSO_3^- + SO_3^{2-}$$
$$S(VI) = H_2SO_{4(g.)} + HSO_4^- + SO_4^{2-}$$

In these equations, dissolved $H_2SO_4$ is assumed to be totally dissociated. Ultimately, S(VI) in droplets ends up deposited at the Earth's surface. In the model, the amount of sulphate deposited is evaluated as a variable. The pH of volcanic water droplets is also a prognostic variable because sulphur species reaction rates and partitioning are pH dependent (Seinfeld and Pandis, 2016). It is dynamically calculated considering the most significant species dissolved in droplets:

$$[H^+] = [HSO_3^-] + 2 \cdot [SO_3^{2-}] + [HSO_4^-] + 2 \cdot [SO4^{2-}] \tag{R1}$$

The main aqueous equilibrium reactions and S(IV) oxidation reactions added to the chemical scheme are summarized in Table:3. The final continuity equation for single $SO_2$ oxidation channels can be expressed as:

$$-\frac{d[SO_2]}{dt} = k_{OH+SO_2} \cdot [SO_2][OH] + \sum_j (k_j \cdot [S(IV)]_{aq}[C_j]_{aq.}) \tag{7}$$

where $k_{OH+SO_2}$ is the rate constant of the gas-phase reaction between OH and $SO_2$ (Sander et al., 2006), $k_j$ the rate constant of the aqueous reaction between $SO_2$ and species $C_j$, whose concentration in the aqueous phase is expressed as $[C_j]_{aq.}$.

Similar continuity equations can easily be derived for all the sulphur species. The continuity equation for atmospheric sulphate S(IV) can be determined by summing all the individual continuity equations of S(IV) species:

$$\frac{d[S(IV)]}{dt} = -k_{OH+SO_2} \cdot [SO_2][OH] - \sum_j (k_j \cdot [S(IV)]_{aq.}[C_j]_{aq.}) - k_d \cdot [SO_{2(aq.)} + HSO_3^- + SO_3^{2-}] - K_{mix} \cdot ([SO_2] - [SO_2]_{(bck)}) \tag{8}$$

where $k_j$ the rate constant of the aqueous reaction between oxidant $C_j$ and relevant [S(IV)] species (see the list of aqueous oxidation reaction in Table:3), and $k_d$ is the deposition coefficient of dissolved sulphur species. Dry deposition as such is not expected to be important in the plume itself compared to wet deposition for our cloudy conditions. Since only wet deposition is considered, only species dissolved in water phases such as aqueous S(IV) ($SO_{2(aq)} + HSO_3^- + SO_3^{2-}$) and S(VI) ($HSO_4^- + SO_4^{2-}$) species are deposited in the model. The deposition is treated as a first order loss with $k_d = 2 \cdot 10^{-6}$ s$^{-1}$, equivalent to a characteristic time scale of 5.7 days (Stevenson et al., 2003b).

The same approach can be used for S(VI) and deposited S(VI):

$$\frac{d[\text{S(VI)}]}{dt} = k_{\text{OH}+\text{SO}_2} \cdot [\text{SO}_2][\text{OH}] + \sum_j (k_j \cdot [\text{S(IV)}]_{aq}[\text{C}_j]_{aq}) - k_d \cdot [\text{HSO}_4^- + \text{SO}_4^{2-}] - K_{mix} \cdot ([\text{S(VI)}] - [\text{S(VI)}]_{(bck)}) \quad (9)$$

$$\frac{d[\text{S(VI)}_{\text{dep}}]}{dt} = k_d \cdot [\text{HSO}_4^- + \text{SO}_4^{2-}] \quad (10)$$

where $\text{S(VI)}_{\text{dep}}$ is the sulphate deposited at the surface.

## 2.4 Oxygen isotope signatures in sulphur oxidation

The mass-balance equation describing the production of S(VI) species is coupled to an oxygen isotope transfer scheme in order to track the evolution of $\Delta^{17}$O in sulphates in water droplets and in sulphates deposited at the surface. Therefore, the specific isotopic anomaly acquired by a S(VI) molecule (produced by the oxidation of a S(IV) molecule by a specific oxidant) is first derived using isotopic transfer equations. New S(VI) isotopes tracers are then created in order to monitor the amount of isotopic anomaly carried out by sulphates in water droplet and deposited at the surface. They are defined as anomaly products ($\Delta^{17}$O $\cdot$ [S(VI)]), and introduced in the model on the basis of the following continuity equation (Morin et al., 2008, 2011):

$$\frac{d}{dt}[\text{S(VI)}] \cdot \Delta^{17}\text{O(S(VI))} = \sum_j [P_j \cdot \Delta^{17}\text{O(S(VI)}_{\text{prod}})_j] - k_d \cdot \Delta^{17}\text{O(S(VI))} \quad (11)$$

where $\Delta^{17}$O(S(VI)) is the isotopic anomaly of atmospheric sulphate, $\Delta^{17}$O(S(VI)$_{\text{prod.}}$)$_j$ is the O-MIF anomaly transferred to sulphate through the specific oxidation channel $j$, and $P_j$ is the oxidation rate of channel $j$. $\Delta^{17}$O(S(VI)$_{\text{prod.}}$)$_j$ is fixed for ozone, $H_2O_2$, and TMI oxidation pathways but it is a prognostic variable for OH (see Table:4).

As deposited sulphate is a variable in the model (S(VI)$_{dep}$), the transfer of isotopic anomaly during deposition is also monitored following a similar equation,

$$\frac{d}{dt}[\text{S(VI)}_{dep}] \cdot \Delta^{17}\text{O(S(VI))} = k_d \cdot [\text{S(VI)}] \cdot \Delta^{17}\text{O(S(VI))} \quad (12)$$

The value of oxygen isotopic anomaly (O-MIF) in sulphate depends on the relative importance of individual $SO_2$ oxidation pathways ($P_j$) and their respective transfer of O-MIF ($\Delta^{17}$O(S(VI))$_j$). Note that the continuity equations of S(VI) and

S(VI)$_{\text{dep}}$ isotopes tracers are integrated with a 4th order Runge-Kutta method algorithm instead of using the CiTTyCAT chemistry solver with the oxidation rates (i.e. P$_j$ in 11) kept constant over a time step (Morin et al., 2007, 2008). This approach allows to keep the chemistry module totally independent from the oxygen isotopic module. The external integration tool does not affect significantly the results. Throughout this study, it is assumed that both SO$_2$ and water vapour (H$_2$O) are not carrying any initial O-MIF. The isotopic composition of magmatic SO$_2$, indeed, follows mass-dependent fractionations and no significant $\Delta^{17}$O, $\Delta^{34}$S and $\Delta^{36}$S have been measured so far (Eiler, 2001). Measurements show that tropospheric H$_2$O does not carry any O-MIF (Uemura et al., 2010), and the same is found for atmospheric SO$_2$ (Holt et al., 1981). It is therefore assumed that the O-MIF found in sulphates only originates from the transmission of isotopic anomaly during the aforementioned reactions of sulphur oxidation.

In order to constrain individual SO$_2$ oxidation pathways from isotopic information, it is first necessary to characterise the specific O-MIFs they transfer to sulphate using isotopic transfer equations.

### 2.4.1 Oxidation by ozone

The few isotopic measurements of tropospheric ozone indicate values of $\Delta^{17}$O (O$_{3,bulk}$) ranging from 20 to 40 ‰ with a mean value of about 25 ‰ (Krankowsky et al., 1995; Johnston and Thiemens, 1997; Thiemens, 2006; Vicars and Savarino, 2014). The location of oxygen isotopes within the structure of ozone is not uniform and heavier isotopes are mostly located at the extremities of the molecule (Janssen, 2005; Bhattacharya et al., 2008). Indeed, molecules that have asymmetrical geometrical structures, and bearing heavier oxygen isotopes on terminal sites, are more energetically stable than their symmetric counterparts (Marcus, 2013). This enrichment in heavy oxygen isotopes at terminal locations of ozone is confirmed by laboratory measurements (Bhattacharya et al., 2008). Ozone does not always react with other molecules via terminal oxygen atoms, although this reaction mechanism is energetically favourable since it requires the breaking of only one molecular bond. During the oxidation of reactive nitrogen leading to production of atmospheric nitrate, most of the oxygen atoms involved in the reaction are from terminal sites (Savarino et al., 2008). Multiple studies found a similar selective reactivity indeed, as during photochemical reactions or for reactions of ozone on solid substrates (Sheppard and Walker, 1983; Bhattacharya et al., 2008). Considering the mean bulk O-MIF and terminal isotopic enrichments, a mean reactive ozone O-MIF ($\Delta^{17}$O (O$_3^*$)) of 36 ‰ has been derived (Bhattacharya et al., 2008; Savarino et al., 2008; Morin et al., 2007). This value is used throughout this study, since it is in accordance with parametrizations used in previous successful model simulations (Michalski et al., 2003; Alexander et al., 2002; Morin et al., 2007; Alexander et al., 2009; Morin et al., 2011).

The value of O-MIF in sulphates generated during the aqueous oxidation by ozone is determined by identifying the origins of each oxygen atom in sulphate during the reaction of oxidation. Ozone transfers to sulphate only one oxygen atom during aqueous sulphur oxidation, while another oxygen atom derive from the water molecule forming aqueous S(IV). The equation describing the transfer of O-MIF to sulphate during oxidation by ozone is:

$$\Delta^{17}O(S(VI))_{O_3+SO_2} = \frac{1}{2} \cdot \Delta^{17}O(SO_2) + \frac{1}{4} \cdot \Delta^{17}O(H_2O) + \frac{1}{4} \cdot \Delta^{17}O(O_3^*) \tag{13}$$

This equation can be simplified because the O-MIF anomalies in $SO_2$ and $H_2O$ are negligible:

$$\Delta^{17}O(S(VI))_{O_3+SO_2} = \frac{1}{4} \cdot \Delta^{17}O(O_3^*) \tag{14}$$

Therefore, the isotopic anomaly in atmospheric sulphates produced in the model during the oxidation of dissolved $SO_2$ through $O_3$ is $\Delta^{17}O(S(VI))_{O_3+SO_2} = 9\,\unicode{x2030}$ (Morin et al., 2007, 2011).

### 2.4.2 Oxidation by hydroxyl radical

In the atmosphere, OH radicals are formed as a result of ozone photolysis in presence of water vapour. In particular, ozone photodissociation can produce an $O^1(D)$ radical, which react with a water molecule to produce two OH radicals. Tropospheric OH radicals are thought not to carry O-MIF anomaly because the exchange of oxygen atoms with water vapour is so fast that it erases any inherited isotopic anomaly in OH. Recall that tropospheric $H_2O$ does not carry any O-MIF because the tropospheric $H_2O$ cycle is entirely controlled by physical processes (condensation, evaporation) and not by chemical processes involving ozone. As a result, the O-MIF signature in OH radicals is expected be zero ($\Delta^{17}O(OH) = 0.0\,\unicode{x2030}$) (Morin et al., 2011). However, when the humidity and hence $H_2O$ levels are very low (e.g. upper troposphere), the rate of isotopic exchange between OH radicals and $H_2O$ molecules decreases so much that freshly produced OH radicals may have time to react with other molecules before losing their isotopic anomaly by isotopic exchange with $H_2O$ (Morin et al., 2007). Under those conditions, when the OH loss reactions and cycling compete with the isotopic exchange with $H_2O$, some of the initial O-MIF originating from ozone is still present in reacting OH. It is also possible for OH loss to compete with the $H_2O$ isotopic exchange when the rate of OH loss is highly enhanced instead of having a reduced rate of $H_2O$ isotopic exchange. This may be the case in volcanic plumes, when $SO_2$ levels are so high that the $SO_2 + OH$ reaction become the dominant chemical loss (Bekki, 1995), accelerating the OH cycling. In order to account for this possibility, instead of assuming a null O-MIF for OH, the O-MIF in the steady-state OH ($\Delta^{17}O(OH)$) is calculated explicitly using the approach developed by Morin et al. (Morin et al., 2007). $\Delta^{17}O(OH)$ is simply derived from the competing balance between the O-MIF erasing isotopic exchange and the total OH loss, typically the reactions with CO and $CH_4$ in the troposphere. Since we consider sulphur-rich volcanic plumes and clouds, the reaction between OH and $SO_2$ is also taken into account.

Considering all the transfers of oxygen atoms, the isotopic mass-balance equation for the OH pathway can be expressed as:

$$\Delta^{17}O(S(VI))_{OH+SO_2} = \frac{1}{2} \cdot \Delta^{17}O(SO_2) + \frac{1}{4} \cdot \Delta^{17}O(OH) + \frac{1}{4} \cdot \Delta^{17}O(H_2O) \tag{15}$$

Since tropospheric $H_2O$ and volcanic $SO_2$ are not thought to carry any O-MIF, the equation can be simplified:

$$\Delta^{17}O(S(VI))_{OH+SO_2} = \frac{1}{4} \cdot \Delta^{17}O(OH) \tag{16}$$

The O-MIF of OH can be derived using the following equation

$$\Delta^{17}O(OH) = x \cdot \Delta^{17}O(OH_{prod.}^*) \tag{17}$$

with

$$\Delta^{17}O(OH^*_{prod.}) = \frac{1}{2} \cdot \Delta^{17}O(O_3{}^*) \tag{18}$$

and

$$x = \frac{D}{D + k^*_{OH+H_2O} \cdot [H_2O]} \tag{19}$$

$$D = k_{OH+CO} \cdot [CO] + k_{OH+CH_4} \cdot [CH_4] + k_{OH+SO_2} \cdot [SO_2] \tag{20}$$

where $k^*_{OH+H_2O}$ is the rate constant for the oxygen atoms exchange reaction between OH and $H_2O$, and $k_{OH+CH_4}$ and $k_{OH+CO}$ are the reaction rate constants for the gas phase reaction of OH with $CH_4$ and CO respectively.

In this approach $x$ represents the competition between the O-MIF erasing effect of isotopic exchange and the O-MIF retaining effect of OH chemical loss; only important loss reactions for tropospheric OH are considered here. $\Delta^{17}O(OH^*_{prod.})$ is the O-

10 MIF of the OH radical freshly produced, and it is assumed that OH is mostly formed by the photolysis of ozone followed by the reaction of $O^1(D)$ with $H_2O$.

The O-MIF in OH ($\Delta^{17}O(OH)$) is determined by this $x$ factor. If OH chemical loss is much faster than the isotopic exchange, $\Delta^{17}O(OH) = 0.5 \cdot \Delta^{17}O(O_3^*)$ (i.e. $x = 1$). If chemical loss is much slower than the isotopic exchange, $\Delta^{17}O(OH) \approx 0‰$ (i.e. $x \ll 1$).

**2.4.3  Oxidation by hydrogen peroxide**

In the troposphere, $H_2O_2$ can quickly dissolve into liquid water phases (Chandler et al., 1988). In a volcanic plume, these phases can be either water droplets or water condensed on solid particles, typically on ash particles. Once in the aqueous phase, $H_2O_2$ oxidizes $SO_2$ by nucleophilic displacement, and its two oxygen atoms are transmitted to the produced sulphate molecule (McArdle and Hoffmann, 1983; Brandt and van Eldik, 1995).

The isotopic balance for the oxidation by $H_2O_2$ in the liquid phase is:

$$\Delta^{17}O(S(VI))_{H_2O_2+SO_2} = \frac{1}{2} \cdot \Delta^{17}O(SO_2) + \frac{1}{2} \cdot \Delta^{17}O(H_2O_2) \tag{21}$$

Since volcanic $SO_2$ is thought to carry no significant O-MIF, the final O-MIF transfer can be simplified:

$$\Delta^{17}O(S(VI))_{H_2O_2+SO_2} = \frac{1}{2} \cdot \Delta^{17}O(H_2O_2) \tag{22}$$

Isotopic measurements of $\Delta^{17}O$ of tropospheric $H_2O_2$ range between 1.30 and 2.20 ‰ with a mean O-MIF of 1.70 ‰

(Savarino and Thiemens, 1999a, b). Using this mean value, sulphate produced by the $H_2O_2$ oxidation is assumed to carry a $\Delta^{17}O(S(VI))_{H_2O_2+SO_2} = 0.87$ ‰ (Savarino et al., 2000).

### 2.4.4 Oxidation by $O_2$/TMI

Isotopic measurements of atmospheric $O_2$ indicate that its O-MIF anomaly is rather small (Luz et al., 1999; Barkan and Luz, 2003). Kinetic isotope fractionation associated to the Dole effect (Dole, 1936) and stratospheric influx of $O_2$ generates a slightly negative O-MIF in tropospheric $O_2$. As theoretical investigations suggest, a slight depletion of $^{17}O$ is indeed found in tropospheric $O_2$, which is accompanied by a slightly negative O-MIF (Barkan and Luz, 2003). Theoretical calculations predict $\Delta^{17}O$ ($O_2$) as low as -0.344 ‰ (Pack et al., 2007) or even, more recently, -0.410 ‰ for tropospheric $O_2$ (Young et al., 2014). Other theoretical calculations suggest a $\Delta^{17}O$ ($O_2$) between 0.141 and -0.305 ‰ (Young et al., 2002).

We assume a $\Delta^{17}O$ ($O_2$) of -0.340 ‰ (Miller, 2002). This value is chosen because it gives a reasonably good agreement between isotopic measurements (Martin et al., 2014) and models (Miller, 2002; Young et al., 2002; Pack et al., 2007). In addition, it has to be kept in mind that there are large uncertainties associated with the exact reaction mechanism of $SO_2$ oxidation catalysed by TMI. We assume that only one oxygen atom of $O_2$ is transmitted to sulphate during the $SO_2$ oxidation (Brandt and van Eldik, 1995; Herrmann et al., 2000).

With these assumptions, the isotopic mass-balance equation for $SO_2$ oxidation by $O_2$/TMI is given by:

$$\Delta^{17}O(S(VI))_{O_2+SO_2} = \frac{3}{4} \cdot \Delta^{17}O(S(IV)) + \frac{1}{4} \cdot \Delta^{17}O(O_2) \tag{23}$$

Since volcanic $SO_2$ is thought to carry no significant O-MIF, we can assume that initial S(IV) species do not carry any O-MIF. Consequently, the isotopic signature associated to this oxidation pathway can be simplified:

$$\Delta^{17}O(S(VI))_{O_2+SO_2} = \frac{1}{4} \cdot \Delta^{17}O(O_2) \tag{24}$$

$\Delta^{17}O$ ($O_2$) being taken as -0.34 ‰ (see above), sulphate produced through this pathway carries a O-MIF ($\Delta^{17}O(S(VI))_{O_2+SO_2}$) almost null, of about -0.09 ‰ (Savarino et al., 2000). The O-MIF signatures of all the S(IV) oxidation pathways used in the model are summarized in Table:4.

## 3 Box model set up

### 3.1 Standard case: initial conditions

All simulations are run for springtime conditions and start at 8.00 a.m. at tropical latitudes (8.3°N). In order to reach stable chemical compositions, notably for medium- and short-lived reactive species, the model is run for 3 days before injecting $SO_2$, then, the evolution of the chemical composition is followed for 7 days. This timescale corresponds approximately to the lifetime of a plume in the free troposphere, in occurrence of low turbulence and low wind shear (Arnold et al., 2007).

Since most of volcanoes are situated in remote areas with their peaks close to the free troposphere, or, at least, with volcanic plumes often ending up in the free troposphere, the environmental conditions are chosen to be representative of the lower free

troposphere with temperature set at 283.15 K, and pressure fixed at 640 mbar (about 3 km altitude). Since we consider cloudy conditions, the relative humidity is set to 100%.

Furthermore, concentrations of reactive species are also set to typical levels found in the tropical lower free troposphere: $O_3$ = 45 ppbv and $H_2O_2$ = 0.1 ppbv. Finally, initial $SO_2$ is set to a mean volcanic plume concentration of 1 ppmv, a value typical of volcanic plumes during degassing (Robock, 2000; Herrmann et al., 2000; Wardell et al., 2004; Mather et al., 2006; Roberts et al., 2012; De Moor et al., 2013; Voigt et al., 2014). The initial pH of the aqueous phase is set to 4.5. It has no impact on the overall model results because the pH is almost immediately driven by $SO_2$ uptake and sulphur oxidation. Preliminary simulations have shown that the initial $SO_2$ concentration is a critical input.

Due to the large amounts of water that can be injected during explosive eruptions, in our simulations it is assumed that volcanic water vapour is largely in excess compared to relative humidity of the free troposphere. Moreover, due to low temperature and pressure of the lower free troposphere, for our simulations it is assumed that volcanic water vapour would mostly condense to produce cloud droplets or to coat ash particles. Therefore, throughout this study relative humidity (RH) inside volcanic plumes is set at 100%, the water saturation point corresponding to the pressure and temperature of the background atmosphere. The LWC (Liquid Water Content) parametrises the amount of liquid water within plumes. High levels of LWC (Liquid Water Content) can be reached, indeed, within volcanic plumes from explosive eruptions. Modelling simulations suggest that LWC as high as $1.6 \ \mathrm{g \, m^{-3}}$ could be reached at the core of water-rich volcanic clouds condensing in the troposphere (Aiuppa et al., 2006b). It is possible that, during the first stages of medium-size eruptions, LWC within the plume could be at least comparable to LWC values of growing cumulus clouds. For all the simulations LWC is set to $1.0 \ \mathrm{g \, m^{-3}}$, a value between experimental measurements (e.g. meteorological clouds) and modelling studies of water-rich volcanic plumes reaching the upper troposphere (Tabazadeh and Turco, 1993; Hoshyaripour et al., 2015). Like $SO_2$, LWC is found to be a critical model input.

TMI concentrations in the liquid phase are set to $[\mathrm{Fe(III)}] = 0{,}5 \ \mu M$ and to $[\mathrm{Mn(II)}] = 0{,}05 \ \mu M$. These values are at the lower end of typical tropospheric measurements with $[\mathrm{Fe(III)}]$ concentrations ranging between 0.5 and 2 μM (Martin, 1984; Martin and Good, 1991; Parazols et al., 2006). Because of uncertainties associated with iron dissolution in volcanic plumes, our TMI concentrations are lower than concentrations found in dust-rich polluted conditions where $[\mathrm{Fe(III)}]$ can reach concentrations of around 5 μM (Herrmann et al., 2000; Parazols et al., 2006). TMI concentrations follow the same relation throughout the whole study and for each simulation $[\mathrm{Mn(II)}] = 0.1 \cdot [\mathrm{Fe(III)}]$ (Martin and Good, 1991).

## 3.2  Model experiments

The objective of the first set of numerical experiments is to assess the competition among oxidation pathways in $SO_2$-rich plumes/clouds for the standard case. Three simulations (S1-S3) are run with oxidation schemes of increasing complexity. They simulate oxidation of $SO_2$: (S1) by OH in gas phase, (S2) by OH in gas phase, and $H_2O_2$ and $O_3$ in aqueous phase, and (S3) by OH in gas phase, and $H_2O_2$, $O_3$ and $O_2$/TMI in aqueous phase.

Since initial $SO_2$ levels, LWC and TMI concentrations in volcanic plumes are relatively uncertain and are key model inputs, the sensitivity of the results to varying conditions within plausible ranges is also explored in additional simulations. Isotopic

anomaly transfers are investigated for atmospheric concentrations stretching from passive degassing/quiescent conditions to sulphur-rich volcanic clouds with varying levels of TMI. The intervals used for the different sensitivity studies are summarised in Table:5.

The first set of sensitivity simulations is devoted to the sensitivity of results to initial $SO_2$ levels in the case of the S1
simulation. It is designed to explore not only the impact of varying $SO_2$ levels on sulphate O-MIF produced by the OH oxidation pathway, but also on OH isotopic signature itself. Recall that the OH isotopic signature ($\Delta^{17}O$ (OH)) is generally assumed zero in the literature (see section 2.4.2).

It is widely recognized that $SO_2$ is the compound emitted by volcanic activity which causes the largest climatic impacts through its conversion into sulphate aerosols (Graf et al., 1998; Robock, 2000; Textor et al., 2004; Langmann, 2014). Emissions
of volcanic $SO_2$ have been measured both in proximity of volcanic vents and in aged plumes. It is possible to constrain a range of concentrations, considering age of plumes and distance from points of emissions. During first stages of plume development concentrations of $SO_2$ in the range of 10-50 ppmv can be reached right in proximity of volcanic vents (Aiuppa et al., 2005, 2006a; Roberts et al., 2012), while concentrations in the range of 0.1-1 ppmv can be found in aged plumes at longer distances from points of emissions (Delmelle, 2003; Carn et al., 2011). These results are confirmed by modelling simulations which
can constrain volcanic emissions by accounting for quick dilution after plume ejection from the vent (Gerlach, 2004; Aiuppa et al., 2006b; Roberts et al., 2009). Consequently, based on atmospheric simulations and on in-situ measurements, the $SO_2$ concentration is set to 1.0 ppmv in the standard case, and is varied from 0.1 to 10 ppmv in the sensitivity simulations.

LWC plays a crucial role in aqueous oxidation of volcanic $SO_2$. The range of LWC considered has been chosen based on LWC observed for different cloud typologies such as mean saturated clouds ($0.1\,g\,m^{-3}$), water-rich cumulus clouds (0.5-1
$g\,m^{-3}$), and cumulonimbus clouds ($1\text{-}2\,g\,m^{-3}$) (Laj et al., 1997; Rosenfeld and Lensky, 1998; Pruppacher et al., 1998; Korolev et al., 2007; Carey et al., 2008). LWC is set to $1\,g\,m^{-3}$ in the standard case and is varied from 0.1 to more extreme values of $2.5\,g\,m^{-3}$ for sensitivity simulations (Tabazadeh and Turco, 1993; Aiuppa et al., 2006b).

Aqueous concentrations of iron ($[Fe(tot)] = [Fe(II)] + [Fe(III)]$) can peak to 9-10 $\mu M$ in the troposphere (Desboeufs et al., 1999, 2001) with $[Fe(III)]$ concentrations between 2.0 $\mu M$ and 5.0 $\mu M$ in polluted conditions if photochemical cycling be-
tween $[Fe(II)]$-$[Fe(III)]$ is inhibited (Parazols et al., 2006). Volcanic eruptions inject large quantities of solid material into the atmosphere in the form of ash. As a result, volcanic plumes/clouds are characterised by high concentrations of ash and minerals (Mather et al., 2003). Ash particles have sizes as large as few mm and they are mainly composed of silica and crystalline minerals of magmatic origin. Glass, olivine, magnetite, hematite and fayelite are among the most common minerals injected during eruptions (Rose and Durant, 2009; Langmann, 2014; Hoshyaripour et al., 2015). These minerals are composed in dif-
ferent proportions by $Fe(II)$ and $Fe(III)$, which are entrapped in the crystalline structure of rocks in different morphologies and compositions. Since large quantities of water are also injected during eruptions, water can condense on mineral particles, especially as the volcanic column reaches higher altitudes and lower temperatures in the troposphere (Tabazadeh and Turco, 1993; Hoshyaripour et al., 2015). Once mineral particles are coated by water, dissolution of iron from the solid mineral surface to the thin liquid water film may take place depending on the acidity of the aqueous phase (Ayris and Delmelle, 2012;
Langmann, 2014; Maters et al., 2016). Acidic conditions (pH < 2.0) due to $H_2SO_4$ condensation or formation within the liquid

phase favour the solubility of minerals containing iron and dissolution of [Fe(III)] (Solmon et al., 2009; Ayris and Delmelle, 2012). Up to a third of total Fe at the ash surface can dissolve into the liquid phase coating volcanic particles (Hoshyaripour et al., 2014) depending on rock composition and gases in the volcanic clouds. Laboratory experiments on dissolution in acidic water of iron from volcanic ashes suggest that [Fe(III)] concentrations of up to $2\,\mu M$ can be reached quickly in the liquid phase when pH reaches $\sim 2$ (Maters et al., 2016). Concentrations as high as [Fe(III)] = $3\,\mu M$ could be reached if pH reaches 1 (Maters et al., 2016). Mobilization of iron ions from ashes could be enhanced for plumes reaching the upper troposphere and undergoing ice formation (Jeong et al., 2012; Shi et al., 2012). High concentrations of [Fe(III)] might persist in the liquid phase depending on the lifetime of water droplets, notably driven by evaporation and condensation cycles (Desboeufs et al., 2001; Langmann, 2014).

Cloud properties are affected by evaporation and condensation cycles changing the pH, the size and number of droplets, while formation of insoluble salts at the surface of ash particles entrapped in cloud droplets can affect mobilization of ions (Ayris and Delmelle, 2012; Langmann, 2014). Therefore, lower acidity combined with the presence of insoluble salts may result in a reduced availability of dissolved TMI in volcanic clouds as the volcanic cloud ages. Over the long term, these conditions can lead to concentration of Fe(III) in water droplets of volcanic clouds which can be lower than typical concentrations found in tropospheric clouds (Desboeufs et al., 1999, 2001). In this study, [Fe(III)] is set to $0.5\,\mathrm{g\,m^{-3}}$ in the standard case and is varied from 0.1 to $3\,\mu M$ in the sensitivity simulations to cover the wide range of possible [Fe(III)] concentrations.

The resulting model $\Delta^{17}O(S(VI))$ (i.e. from standard and sensitivity simulations) are compared to sulphate O-MIF found in tropospheric volcanic sulphates extracted from ash-deposits of small and medium-size tropospheric explosive eruptions of the present geological era (Bindeman et al., 2007; Martin et al., 2014; Bao, 2015), or in sulphate aerosols collected at volcanic vents, most certainly primary sulphate (Mather et al., 2006).

## 4 Results and discussion

### 4.1 Isotopic constraints on individual oxidation pathways of volcanic $SO_2$

#### 4.1.1 Gaseous oxidation by OH

S1 simulates the O-MIF transfer to sulphate in the absence of aqueous oxidation for standard conditions; $H_2SO_4$ is only produced by the reaction between hydroxyl radicals and $SO_2$ in the gas phase. As soon as $SO_2$ is injected, it reacts with OH to produce gaseous $H_2SO_4$. Fig.:2 shows the decay of $SO_2$ levels from 1.5 ppmv to 1.26 ppmv after 7 days. As expected, gas-phase concentrations of other $SO_2$ oxidants (i.e. $O_3$ and $H_2O_2$) are not substantially affected by the $SO_2$ injection. Since $H_2SO_4$ is very soluble, once produced in the gas phase, it ends up dissolved in the liquid phase where it shifts the pH towards acidic values because of its quasi-complete dissociation. The concentration of atmospheric sulphate (S(VI)) is driven by the gas-phase production and deposition. Sulphate production follows a diurnal cycle because of the diurnal production of OH. There is no significant production of S(VI) during night-time. Fig.:3 shows the time evolutions of the liquid phase pH, of the atmospheric sulphate O-MIF ( $\Delta^{17}O(S(VI))$ ), and of deposited sulphate O-MIF ( $\Delta^{17}O(S(VI))_{dep}$ ) following the injection of

SO$_2$. The pH exhibits diurnal variations because sulphate is only produced during daytime. After an initial spike around +0.95 ‰, atmospheric sulphate O-MIF ($\Delta^{17}$O(S(VI))) declines very slowly to +0.9 ‰ after 7 days. Since SO$_2$ is only oxidized by OH, the evolution of $\Delta^{17}$O(S(VI)) reflects the evolution of the OH isotopic signature ($\Delta^{17}$O (OH)). Unexpectedly, it is found to be very different from zero. Usually, the exchange of oxygen atoms with water vapour is so fast in the troposphere that it erases any inherited isotopic anomaly in OH. However, in our standard conditions, SO$_2$ levels are so high that the SO$_2$ + OH reaction become the overwhelmingly dominant loss (Bekki, 1995), accelerating greatly the OH cycling. Under those conditions, the OH loss is so enhanced that the reaction competes with the isotopic exchange with H$_2$O. Therefore OH radicals can react before their isotopic anomaly is entirely erased by the isotopic exchange, and they maintain a significant positive signature. As SO$_2$ concentration decreases slowly, the rate of OH loss decreases and, as expected, so is $\Delta^{17}$O(S(VI)). The evolution of deposited sulphate O-MIF somewhat follows the evolution of the atmospheric sulphate O-MIF but with the time lag which is related to the characteristic timescale of the atmospheric sulphate deposition specified in the model (5.7 days). The final O-MIF of deposited sulphate (resulting from the cumulative effect of sulphate deposited since the SO$_2$ injection) is 0.92 ‰, which is distinctively higher than most measurements from volcanic sulphate sampled in volcanic ash-deposits (see Table:1).

### 4.1.2   Isotopic signature of OH: dependence on initial SO$_2$

Since $\Delta^{17}$O(OH) is sensitive to the SO$_2$ level, additional S1 simulations are carried out with SO$_2$ concentration differing by 3 orders of magnitude. Fig.:4 shows the time evolution of OH and sulphate O-MIF for two different initial SO$_2$ concentrations (i.e. 1 ppbv and 1 ppmv). The upper plot shows the time evolution of OH for the two SO$_2$ cases. As soon as the SO$_2$ is injected, OH concentration drops sharply in the high SO$_2$ case, whereas it remains unaffected in the low SO$_2$ case. At such high SO$_2$ concentration, the reaction between OH and SO$_2$ becomes the main OH loss. As a result, OH concentration and lifetime drop and its cycling is much faster. The lower plot of Fig.:4 illustrates the effect on the value of O-MIF transferred to sulphate via OH oxidation. For 1 ppbv of initial SO$_2$ instead of 1 ppmv, $\Delta^{17}$O(OH) is negligible (of the order of 0.01 ‰), thus the sulphate O-MIF is quasi-zero. For 1 ppmv of initial SO$_2$, shortly after the injection, the sulphate produced has an isotopic signature of 0.7 ‰. It then declines slowly as the SO$_2$ concentration decays slowly. In Eq.:17, the competition between OH loss channels and the isotopic exchange with water is represented by the $x$ ratio. In the simulation with 1 ppmv of initial SO$_2$, $\Delta^{17}$O(OH) decreases from 10.8 to 7.6 ‰, whereas the variation is very small in the simulation with 1 ppbv of initial SO$_2$.These results suggest that OH could have a positive O-MIF in volcanic sulphur-rich plumes and clouds which is subsequently transferred to the produced sulphate. Since most isotopic measurements indicate that O-MIF in volcanic sulphate is very close to zero, at least within the measurement errors of about 0.1 ‰ typically, other oxidation pathways that are mostly mass-dependent (i.e. null O-MIF) have to contribute very significantly to the formation of volcanic sulphate. The other known oxidation pathways of SO$_2$ are heterogeneous.

### 4.1.3  Gaseous and heterogeneous oxidation by $O_3$ and $H_2O_2$

The S2 simulation is the same as S1 except that it also includes the aqueous oxidation of $SO_2$ by $H_2O_2$ and $O_3$. Fig.:5 shows evolving concentrations of atmospheric species as oxidation takes place. Almost as soon as $SO_2$ is injected, the $H_2O_2$ concentration drops from about 20 ppbv to less than a pptv. At the same time, as in S1, the pH quickly drops to less than 3 (see Fig:6). Just after the injection, $H_2O_2$ is initially the overwhelmingly dominant oxidant (Martin et al., 2014). The contribution of oxidation by $O_3$ is almost negligible under acidic conditions and the oxidation by OH in the gas phase is much slower initially that aqueous oxidation by $H_2O_2$. However, as the $SO_2$ concentration vastly exceeds the $H_2O_2$ concentration (by 3 orders of magnitude), $H_2O_2$ is very quickly consumed by reaction with $SO_2$ in the liquid phase; recall for each molecule of $SO_2$ oxidized by $H_2O_2$, one molecule of $H_2O_2$ is consumed. The difference in concentration between $SO_2$ and $H_2O_2$ is such that, as soon as a molecule of $H_2O_2$ enters the liquid phase, it is consumed. The sharp drop in $H_2O_2$ concentration is thus limited by $H_2O_2$ gas phase diffusion to the surface of the liquid phase, which is followed by its quick reaction with $S(IV)$. After the initial drop, the $H_2O_2$ concentration stays very low with very large diurnal variations. The daytime concentration approaches pptv levels because the loss to the liquid phase is balanced by gas-phase photochemical production. After the drop in $H_2O_2$, most of the $SO_2$ is oxidised by OH in the gas-phase. As shown in Fig:6, the pH in S2 follows a trend similar to the one in S1. The sulphate O-MIF in S2 is higher than in S1. In the first phase, sulphate is produced with a rather high O-MIF signature because the contribution of $O_3$ to $SO_2$ oxidation is significant with an initial pH set to 4.5 (i.e. as high as 50% of $SO_2$ is oxidised by $O_3$ within the first hours). The O-MIF of produced sulphate ($\Delta^{17}O(S(VI)_{PRD})$) peaks early on at 4 ‰. However, the pH drops very quickly as more $S(VI)$ is produced in the aqueous phase. As a result, the pH-dependent oxidation rate by $O_3$ decreases quickly and hence so does $\Delta^{17}O(S(VI)_{PRD})$. $H_2O_2$ is completely consumed within 15 minutes during the first timesteps, and it does not contribute to the $SO_2$ oxidation thereafter. The oxidation of $SO_2$ is dominated by OH except during the early phase. The final O-MIF in deposited sulphate is 1.1 ‰, originating mostly from OH oxidation. Recall that, when OH is generated via its main production channel, it carries an isotopic anomaly. Under common (non-volcanic) conditions, the OH anomaly is so rapidly erased by isotopic exchange with $H_2O$ that, when OH reacts, it carries no anomaly. However, when $SO_2$ levels are very high, OH might react very quickly with $SO_2$ without having lost its anomaly by isotopic exchange. In this situation, the value of $\Delta^{17}O(OH)$ is determined by the competition between the $SO_2 + OH$ reaction and the OH isotopic exchange with $H_2O$. As $SO_2$ concentration decays with time, $\Delta^{17}O(OH)$ decreases because the $SO_2 + OH$ reaction slows down and becomes less competitive with respect to the isotopic exchange (see Eq.:15-20). This explains why $\Delta^{17}O(OH)$ decreases from 4 ‰ to roughly 3 ‰ by the end of simulation, resulting in produced sulphates with respectively O-MIF of 1 and 0.75 ‰. The value of O-MIF on deposited sulphates produced in this simulation is still much higher than most O-MIF measurements in tropospheric volcanic sulphates (see Table:1). In order to produce mass-dependent sulphates without O-MIF ($\Delta^{17}O= 0.0\pm0.1$‰), another oxidant needs to be dominant and it has to carry a small or null O-MIF anomaly.

### 4.1.4 Gaseous and heterogeneous oxidation by $O_3$, $H_2O_2$ and $O_2$/TMI

Simulation S3 includes all the major pathways of oxidation involved during formation of sulphate in $SO_2$-rich clouds (i.e. without significant halogens concentrations compared to sulphur species). Fig.:7 shows the evolution of the chemical species concentrations. In S3, $H_2O_2$ is very quickly depleted just after the $SO_2$ injection as in S2. However, there is much less $SO_2$ left at the end of the run in S3 than in S2 and S1 and conversely there is more $S(VI)$ produced in S3 than in S2 and S1. With the TMI catalysed oxidation added to the S2 chemical scheme, heterogeneous chemistry becomes competitive with the gas-phase oxidation by OH and lead to faster formation of $S(VI)$. The pH is also lower (see Fig.:8), confirming that more sulphates are in aqueous solution. The final O-MIF in deposited sulphates is about 0.3 ‰. This value is lower than the values calculated in simulations S1 and S2 and closer but still higher than the range of isotopic measurements carried out on volcanic sulphate (see Table:1). This result suggests that heterogeneous $SO_2$ oxidation by $O_2$/TMI is the only pathway able to explain sulphate isotopic measurements with the current chemical scheme. Sensitivity studies are however needed to test the responses of the system to varying conditions of heterogeneous oxidation. Consequently, we conduct further simulations to probe the effects of LWC and TMI aqueous concentrations on the final MIF in deposited sulphate and assess the robustness of the overall results.

## 4.2 Sensitivity studies

### 4.2.1 Influence of $SO_2$ on sulphate O-MIF

In the first set of sensitivity simulations (Z1), the response of the system to various concentrations of $SO_2$ is tested for the standard conditions with all the oxidation channels included in the model. Fig:9 shows the time evolution of O-MIF in the produced sulphate for different initial $SO_2$ concentrations. The initial concentration of volcanic $SO_2$ is varied from 10 ppbv to 10.0 ppmv. LWC is set to $1.0\,\mathrm{g\,m^{-3}}$ and $[\mathrm{Fe(III)}] = 0.5\,\mu\mathrm{M}$. The response of the system and, in particular, of produced sulphate O-MIF to varying $SO_2$ levels is complex and not at all linear. In Table:6 the different contributions of the oxidation pathways are reported for different initial $SO_2$ concentrations, in function of: O-MIF in OH, O-MIF in deposited sulphate. The isotopic anomalies and the contribution of different pathways of oxidation are reported for one day after sulphur injection, and for the end of the simulations. For an initial $SO_2$ of 10 ppbv, $H_2O_2$ is the dominant pathway of sulphur oxidation with the OH oxidation pathway representing about a third of the total; the final O-MIF in the deposited sulphate is 0.50 ‰. When the initial $SO_2$ concentration is increased (from 10 ppbv to 30 and then 100 ppbv), sulphate O-MIF decreases because the $H_2O_2$ contribution to $SO_2$ oxidation drops whereas the OH contribution becomes dominant. The drop in the $H_2O_2$ contribution is mostly due to the increased acidity of cloud droplets at higher $SO_2$ concentrations, resulting in much reduced uptake of $SO_2$ combined to a much smaller fraction of aqueous $S(IV)$ in the form of $HSO_3^-$, the reactant for the $S(IV)$ oxidation by $H_2O_2$ (McArdle and Hoffmann, 1983). Above 100 ppbv, instead of decreasing, sulphate O-MIF increases at higher initial $SO_2$ concentration, with 10 ppmv being the maximum $SO_2$ concentration considered here. This inversion in the evolution of $\Delta^{17}O(S(VI))$ with increasing $SO_2$ concentration originates from the change in $\Delta^{17}O(OH)$. Up to 100 ppbv of $SO_2$, $\Delta^{17}O(OH)$ is more or less negligible and consequently the OH oxidation produces sulphate with insignificant O-MIF. However, at 300 ppbv of initial $SO_2$, $\Delta^{17}O(OH)$ is equal to 0.8 ‰ at the start of simulation and is still greater than 0.3 ‰ at the end of the run. The higher

SO$_2$ concentration is, the higher $\Delta^{17}$O(OH) is. At 10 ppmv of SO$_2$, $\Delta^{17}$O(OH) = 11.5 ‰. Recall that the maximum possible value of $\Delta^{17}$O(OH) is 18.0 ‰, corresponding to conditions where the rate of the isotopic exchange is negligible compared to the rate of OH chemical loss. Interestingly, the overall contribution of OH to the sulphur oxidation peaks at 69% for initial SO$_2$ equal to 300 ppbv. At higher concentrations, the OH contribution decreases reaching 33% for the simulation with 10 ppmv of initial SO$_2$. At the same time, the contribution of O$_2$/TMI oxidation (the only channel with a negligible O-MIF signature) increases sharply becoming even dominant (58%) for the simulation with 10 ppmv of initial SO$_2$. Nonetheless, the very large increase in $\Delta^{17}$O(OH) is the main driver of $\Delta^{17}$O(S(VI)) for high SO$_2$ levels, because the OH contribution remains important even for the simulation with 10 ppmv of initial SO$_2$. Note that the H$_2$O$_2$ contribution declines all the way with increasing initial SO$_2$ (from 10 ppbv to 10 pmmv) which is the inverse of the O$_2$/TMI contribution evolution.

Overall, the model-calculated contribution of the OH pathway to volcanic sulphur oxidation does not drop below 30% for the standard conditions considered here. As a result, the O-MIF of deposited sulphate O-MIF is expected to depend strongly on the amount of SO$_2$ injected, via the dependency of OH isotopic signature on SO$_2$ concentration. Since volcanic SO$_2$ usually reaches ppmv levels during the first stages of volcanic plume (Roberts et al., 2009, 2012; Oppenheimer et al., 2013; Voigt et al., 2014), our results suggest that volcanic sulphate should carry positive O-MIF anomalies that exceed isotopic measurements uncertainties ($\approx 0.1$ ‰). This is not supported by atmospheric measurements of volcanic sulphate isotopic composition which mostly lie close to zero within measurements uncertainties. Other environmental conditions or reaction mechanisms have to be considered to explain the lack of O-MIF in volcanic sulphate

### 4.2.2 Influence of LWC on sulphate O-MIF

The second set of sensitivity runs (Z2) tests the influence of the LWC amount on model results, notably the final sulphate O-MIF. Fe(III) aqueous concentration is fixed to 0.5 µM, and the initial concentration of SO$_2$ is set to 1.5 ppmv. Fig:10 shows the evolution of $\Delta^{17}$O(S(VI), DEP) for varying LWC. Unlike the influence of SO$_2$, sulphate O-MIF is a monotonic function of LWC. The higher the LWC is, the lower the sulphate O-MIF is. Higher values of LWC favours higher dissolution rate of SO$_2$ in droplets and push the dynamics of oxidation towards liquid phase reactions. Also, at high LWC, H$_2$O$_2$ is more quickly depleted from the gas-phase because of faster uptake in the liquid phase. Overall, higher LWC favours the O$_2$/TMI oxidation pathway since lower acid concentration promotes the dissolution of SO$_2$ in the aqueous phase. Higher LWC does not directly affect the vapour pressure of the system, because it is assumed that cloud droplets are formed at high water saturation. Therefore, changes in cloud LWC do not affect the the isotopic composition of produced OH. For the range of LWC considered here (from 0.1 to 3 g m$^{-3}$), the final O-MIF of deposited sulphate varies from 0.8 to 0.2 ‰. The results show that, in volcanic clouds and plumes, sulphate O-MIF is affected by the LWC value. However, high LWCs alone do not appear to be sufficient to reproduce most of the isotopic measurements in volcanic sulphate.

### 4.2.3 Influence of TMI on sulphate O-MIF

The third and last set of sensitivity runs (Z3) tests the influence of TMI concentrations on model results, notably the final sulphate O-MIF. Laboratory experiments on iron mobilization from ash indicate that an average concentrations of [TMI] =

3 μM could be reached within the first hour of ash exposure to very acidic water in the case of silica ashes (Maters et al., 2017). According to the measurements, concentration of dissolved [Fe(III)] generally varies in the range of 0.1 to 2 μM depending on the mineral composition (Hoshyaripour et al., 2015; Maters et al., 2016, 2017). However, there is a lot of uncertainty on the typical concentrations of dissolved iron mobilized in volcanic plumes. In Fig:11 the plot shows the evolution

of deposited sulphate O-MIF to varying concentrations of dissolved TMI (from 0.1 to 3 μM). At relatively low concentration of dissolved Fe(III) (0.1 - 1.0 μM), deposited sulphates are generated with high O-MIFs and a final O-MIF greater than 0.4 ‰. At higher concentrations of [Fe(III)] = 2 or 3 μM, deposited sulphate O-MIF reaches values as low as 0.25 and 0.15 ‰ respectively. According to the laboratory experiments, these high Fe(III) concentrations require a pH below pH $\leq$ 2, enhancing the mobilization of [Fe(III)]. In the simulations, the pH is close to this threshold value (see Fig.:8). In conclusion, the

model simulations suggest that deposited sulphate with very low O-MIF values, consistent with most isotopic measurements of volcanic secondary sulphate (less than 0.1 ‰), can only be achieved with highly enhanced Fe(III) mobilization.

## 5 Conclusions

We use the tropospheric photochemical box-model CiTTyCAT to analyse why most oxygen isotopic measurements of tropospheric volcanic sulphate indicate that volcanic sulphates are essentially mass-dependent (i.e. O-MIF anomalies lying close to

15 zero within measurements uncertainties of $\pm0.1$ ‰ typically). This is also observed for volcanic sulphate collected very far from volcanoes where secondary sulphate (produced by oxidation of volcanic sulphur precursors, mostly $SO_2$) is expected to vastly dominate. This lack of O-MIF in volcanic sulphate is rather intriguing because secondary sulphates originating from other sources exhibit significant O-MIF. A major difference between volcanic sulphur and other sources is that it is often emitted within very dense volcanic plumes whose chemical compositions are radically different from background air. The chemical

environment of the plumes may affect the oxidation pathways and hence sulphate isotopic composition.

A new sulphur isotopic O-MIF scheme is implemented in the model in order to monitor the transfer of O-MIF from oxidants to sulphate during the oxidation of volcanic $SO_2$. A range of simulations are performed in order to explore in details the different pathways of $SO_2$ oxidation (gas-phase oxidation by OH and aqueous oxidation by $O_3$, $H_2O_2$ and $O_2$/TMI) and, more importantly for O-MIF, their relative importance for a range of possible volcanic conditions. The first salient finding is

25 that, according to the model calculations, OH should carry a very significant O-MIF in sulphur-rich volcanic plumes. This implies that, when volcanic sulphate is produced in the gas phase via $SO_2$ oxidation by OH, its O-MIF should have a very significant positive value. Since most isotopic measurements of volcanic sulphate do not indicate the presence of O-MIF, the OH oxidation pathway cannot be the dominant channel for volcanic sulphur. Nonetheless, uncertainties on the rate constant of the isotopic exchange between OH and $H_2O$ (Dubey et al., 1997) and, more generally, on photochemical modelling are

30 substantial (Ridley et al., 2017). It would be useful for this unexpected model predictions of O-MIF in OH, and hence volcanic sulphate produced in gas phase, to be tested in a controlled environment, ideally laboratory experiments of $SO_2$ oxidation with a well constrained OH chemical budged, notably the loss processes. The second important finding from these simulations is that, although $H_2O_2$ is a major oxidant of $SO_2$ throughout the troposphere, it is very rapidly consumed in sulphur-rich volcanic

plumes. Since $H_2O_2$ produced within the plume and the entrainment of $H_2O_2$ from the atmospheric background represent also relatively weak sources, $H_2O_2$ is found to be a minor oxidant for volcanic $SO_2$ whatever the liquid water content. According to the simulations, oxidation of $SO_2$ by $O_3$ is negligible because volcanic aqueous phases are too acidic. The model predictions of minor or negligible sulphur oxidation by $H_2O_2$ and $O_3$, two oxidants carrying large O-MIF, are consistent with the lack of O-MIF seen in isotopic measurements of volcanic tropospheric sulphate. The third finding is that oxidation by $O_2$/TMI in volcanic plumes could be very substantial and, in some cases, dominant, notably because the rates of $SO_2$ oxidation by OH, $H_2O_2$, and $O_3$ are vastly reduced in a volcanic plume compared to the background air. Only cases where sulphur oxidation by $O_2$/TMI is very dominant can explain the isotopic measurements of volcanic tropospheric sulphate. We stress that oxidation by $O_2$/TMI is poorly constrained in model simulations because of the lack of measurements of TMI aqueous concentrations in volcanic plumes. It is worth pointing out that our model results are only applicable to cloudy volcanic plumes. Nonetheless, water clouds do not always form in volcanic plumes, notably during passive degassing. It would be interesting to also consider cloud-free plumes where the condensed phase is concentrated sulphuric acid within sulphate aerosols. In particular, these particles have a chemical reactivity radically different from water droplets.

A potentially significant limitation of the model simulations is the omission of volcanic halogens. Indeed, volcanic halogens are known to undergo multi-phase chemistry, resulting in ozone depletion and possibly impacting the oxidation of volcanic $SO_2$ (Bobrowski et al., 2003; Bobrowski and Platt, 2007; Millard et al., 2006; Bobrowski et al., 2007; Roberts et al., 2009; von Glasow, 2010; Roberts et al., 2014; Jourdain et al., 2016). Halogen species such as HOBr may also directly oxidise $SO_2$ in the aqueous phase (Chen et al., 2017), but this oxidation pathway has not been quantified yet for volcanic plumes. Overall, the present simulations might only be representative of degassing or eruptions with extremely low halogen emissions, typically originating from intraplate and rift volcanic activity. It is certainly worth exploring the potential impact of halogens in the case of halogen-rich eruptions, notably for volcanic plumes where water does not condense and hence only sulphate aerosols are present. Since the heterogeneous conversion of halogen halides into radicals is known to be fast on sulphate aerosols (Bobrowski et al., 2007; Roberts et al., 2009; von Glasow, 2010; von Glasow and Crutzen, 2013), halogens might impact significantly the plume chemistry and the isotopic composition of secondary sulphate for halogens rich conditions.

*Code and data availability.* galeazzo.tommaso@latmos.ipsl.fr; tommaso.galeazzo@gmail.com

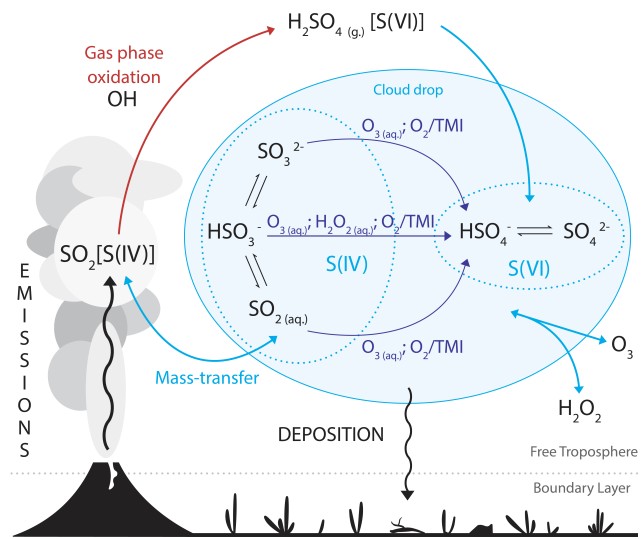

**Figure 1.** Diagram of the sulphur scheme implemented in CiTTyCAT.

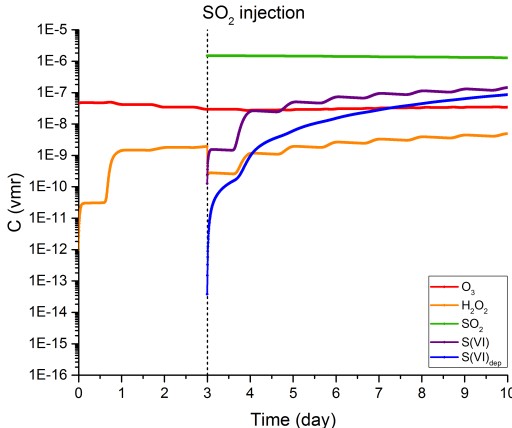

**Figure 2.** Evolution of the gas-phase concentrations of atmospheric species during the S1 simulation (see text). The simulation starts at 8:00 a.m. and $SO_2$ is injected after 3 days. During S1 simulation the concentration of injected $SO_2$ drops from 1.5 ppmv to a final value of 1.27 ppmv.

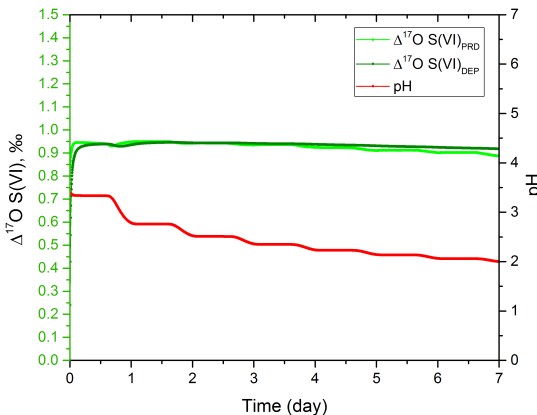

**Figure 3.** Time evolution of $\Delta^{17}O(S(VI))$ in produced and deposited, and of the pH of the liquid phases in volcanic plumes during simulations S1, following injection of $SO_2$ in the box. The change of pH in water droplets is also reported as a function of time.

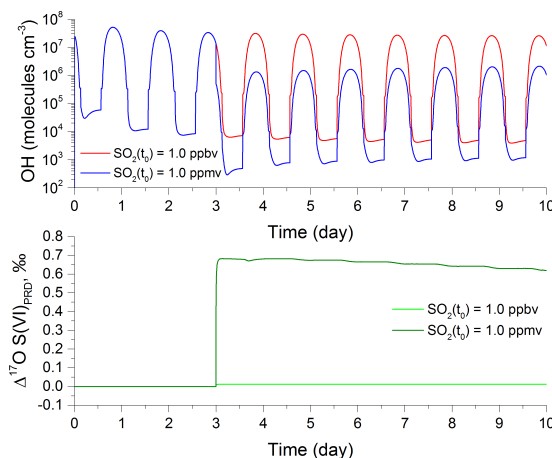

**Figure 4.** time evolution of the O-MIF transfer from OH to $H_2SO_{4(g)}$ at two different initial concentrations of $SO_2$. The light green line represents initial concentration of $S(IV)$ = 1 ppbv (e.g. mean troposphere); the dark green line represents an initial concentration of $SO_2$ = 1 ppmv (e.g. volcanic plumes/clouds). The upper figure shows concentration trends for OH during the two different scenarios.

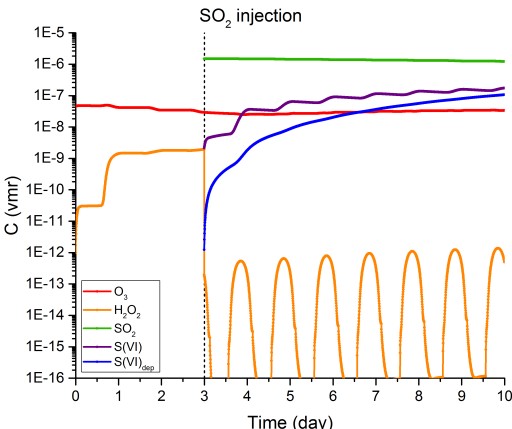

**Figure 5.** Time evolution in the gas-phase concentrations of $SO_2$, its tropospheric oxidants and produced and deposited sulphates during S2. During S2 simulation the concentration of injected $SO_2$ drops from 1.5 ppmv to a final value of 1.2 ppmv.

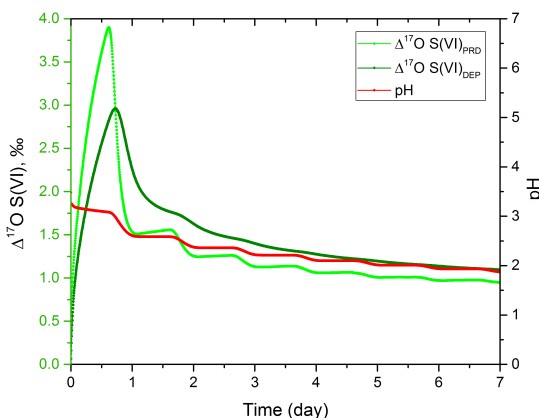

**Figure 6.** Temporal evolution of $\Delta^{17}O(S(VI))$ in produced and deposited sulphates, and of pH during the S2 simulation.

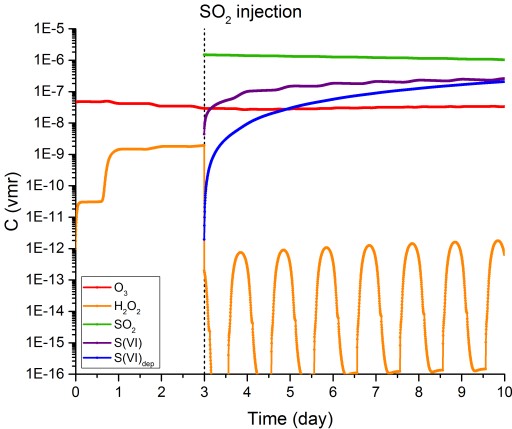

**Figure 7.** Time evolution in the gas-phase concentrations of $SO_2$, its tropospheric oxidants and produced and deposited sulphates during S3. During S3 simulation the concentration of injected $SO_2$ drops from 1.5 ppmv to a final value of 1 ppmv.

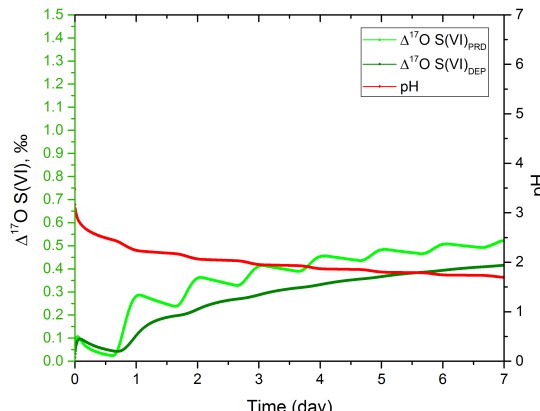

**Figure 8.** Time evolution of $\Delta^{17}O(S(VI))$ in produced and deposited sulphates, and of pH of the liquid phases of volcanic plumes during simulation S3.

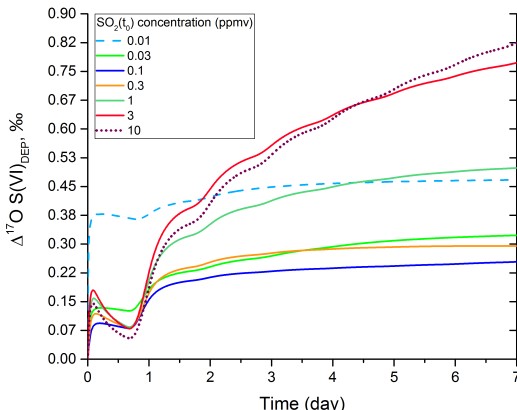

**Figure 9.** Temporal evolution of $\Delta^{17}O(S(VI))_{dep}$ at different initial concentrations of $SO_2$. The dashed line represents simulation where $H_2O_2$ is the major $SO_2$ oxidant, straight lines are simulations for which OH is the major oxidant, and dot lines are simulations for which $O_2/TMI$ is the major pathway of oxidation. The equivalent pathways contributions are summarised in Table:6. Other critical parameters are set to: LWC = $1.0 \ g \ m^{-3}$ and $[Fe(III)] = 0.5 \ \mu M$

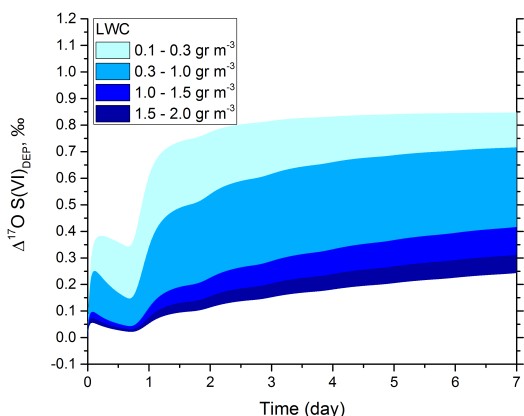

**Figure 10.** Temporal evolution of $\Delta^{17}O(S(VI))_{dep}$ at different values of liquid water content. Other initial critical parameters are set to: $[Fe(III)] = 0.5\,\mu M$ and $[SO_2]_0 = 1.5$ ppmv.

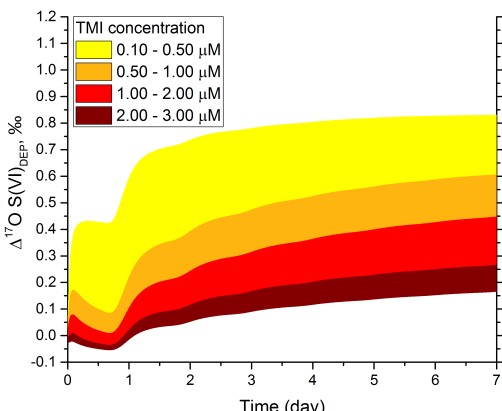

**Figure 11.** Temporal evolution of $\Delta^{17}O(S(VI))_{dep}$ at different concentrations of TMI in aqueous solution. Other initial parameters were set: LWC = 1.0 g m$^{-3}$ and [SO$_2$]$_0$ = 1.5 ppmv.

**Table 1.** Oxygen isotopic composition of volcanic sulphates from different tropospheric emissions of the present geological era.

| Volcano & Date of Eruption | Sample distance (km) | Source | $\Delta^{17}O$ (‰) | Reference |
|---|---|---|---|---|
| **Popocatépetl** (Mexico), 2008 | 25 | ash | 0.35 | (Martin et al., 2014) |
| **Spurr** (Alaska, USA), 1992 | 265 | ash | -0.14 | (Martin et al., 2014) |
| **Fuego** (Guatemala), 1974 | 57 | ash | -0.04 | (Martin et al., 2014) |
| **Negro Cerro** (Nicaragua), 1947 | 12 | ash | -0.06 | (Martin et al., 2014) |
| **Parícutin** (Mexico), 1948 | 5 | ash | 0.13 | (Martin et al., 2014) |
| **Mt. St. Helens** (USA), 1980 | 400 | ash | 0.02 | (Martin et al., 2014) |
| **Gjálp** (Iceland), 1998 | < 30 | ash | -0.07 | (Martin et al., 2014) |
| **Pinatubo** (Philippines), 1991 | < 50 | ash | -0.04 | (Bindeman et al., 2007) |
| **Pinatubo** (Philippines), 1991 | < 50 | ash | 0.19 | (Bindeman et al., 2007) |
| **Spurr** (USA), 1953 | n.a. | ash | 0.06 | (Bindeman et al., 2007) |
| **Vesuvius** (Italy), 1872 | n.a. | ash | -0.07 | (Bao et al., 2003) |
| **Popocatépetl** (Mexico), 1997 | n.a. | ash | -0.08 | (Bao et al., 2003) |
| **Spurr** (USA), 1992 | n.a. | ash | 0.06 | (Bao et al., 2003) |
| **Fuego** (Guatemala), 1974 | 55 | ash | -0.03 | (Bao et al., 2003) |
| **Pinatubo** (Philippines), 1991 | n.a. | anhydrite from pumice | -0.01 | (Bao et al., 2003) |
| **Santorini** (Greece), Minoan age | n.a. | pumice + ash | 0.09 | (Bao et al., 2003) |
| **Masaya** (Nicaragua), 2003 | 0 | aerosols | 0.1 | (Mather et al., 2006) |
| **Masaya** (Nicaragua), 2003 | 0 | aerosols | 0.2 | (Mather et al., 2006) |

[*] Refer to (Martin, 2018) for a more extensive description regarding oxygen isotopic anomalies measured in tropospheric volcanic sulphate of present and past geological eras.

**Table 2.** Sulphur aqueous equilibria

| Equilibrium | K ($M^{-1}$), | $k_{298(forward)}$ ($M^{-1}s^{-1}$), | $E_a/R$ (K), | $k_{298(back)}$ ($M^{-2}s^{-1}$), |
|---|---|---|---|---|
| $SO_{2(aq.)} + H_2O \rightleftharpoons HSO_3^- + H_3O^+$ | $3.13 \cdot 10^{-4}$ | $6.27 \cdot 10^4$ | -1940 | $2 \cdot 10^8$ [a,c] |
| $HSO_3^- + H_2O \rightleftharpoons SO_3^{2-} + H_3O^+$ | $6.22 \cdot 10^{-8}$ | 3110 | -1960 | $5 \cdot 10^{10}$ [a,c] |
| $H_2SO_4 + H_2O \rightarrow HSO_4^- + H_3O^+$ | | $\infty$ | | |
| $HSO_4^- + H_2O \rightleftharpoons SO_4^{2-} + H_3O^+$ | $1.02 \cdot 10^{-2}$ | $1.02 \cdot 10^9$ | -2700 | $1 \cdot 10^{11}$ [b,c] |

[a] (Beilke and Gravenhorst, 1978); [b] (Redlich, 1946); [c] (Graedel and Weschler, 1981)

**Table 3.** Sulphur chemistry scheme

| Gaseous reaction | k | units |
|---|---|---|
| $SO_2 + OH + M \rightarrow HOSO_2 + M$ | $4.62 \cdot 10^{-31} \cdot (T/298.0)^{-3.90}$ | $cm^6 molecule^{-2}s^{-1}$ [a] |
| $HOSO_2 + O_2 \rightarrow HO_2 + SO_3$ | $1.30 \cdot 10^{-12} \cdot (-330/T)^{-3.90}$ | $cm^3 molecule^{-1}s^{-1}$ [a] |
| $SO_3 + H_2O \rightarrow H_2SO_4$ | $9.10 \cdot 10^{-13}$ | $cm^3 molecule^{-1}s^{-1}$ [a] |

| Aqueous reaction | k($aq$) | units; (T) |
|---|---|---|
| $SO_{2(aq.)} + O_3 \rightarrow S(VI) + O_2$ | $2.4 \cdot 10^4$ | $Ms^{-1}$ [b] |
| $HSO_3^- + O_3 \rightarrow S(VI) + O_2$ | $3.7 \cdot 10^5$ | $Ms^{-1}$ [b] |
| $SO_3^{2-} + O_3 \rightarrow S(VI) + O_2$ | $1.5 \cdot 10^9$ | $Ms^{-1}$ [b] |
| $HSO_3^- + H_2O_2 \rightarrow S(VI) + H_2O$ | $\dfrac{k_{H_2O_2} \cdot [H^+]}{1 + K_{(eq.)} \cdot [H^+]}$ | $Ms^{-1}$ [b] |
| | with $K_{(eq.)} = 13$ | $M^{-1}$ [b] |
| | and $k_{H_2O_2} = 7.5 \cdot 10^7$ | $M^{-2}s^{-1}$ [b] |
| $S(IV) + \dfrac{1}{2} O_2 \xrightarrow{TMI} S(VI)$ | $750 \cdot [Mn(II)] + 2600 \cdot [Fe(III)] + 1.0 \cdot 10^{10}[Mn(II)][Fe(III)]$ | $s^{-1}$ [c] |

[a] (Atkinson et al., 2004); [b] (Hoffmann, 1986); [c] (Martin and Good, 1991)

**Table 4.** O-MIF signatures of S(IV) oxidation pathways in the model

| Oxidant | O-MIF pathway (‰) |
|---------|-------------------|
| OH | calculated (0 to a maximum of 4.5) |
| $H_2O_2$ | 0.87 |
| $O_3$ | 9 |
| $O_2/TMI$ | -0.09 |

**Table 5.** Ranges of $SO_2$, LWC and TMI explored in the sensitivity studies

| | |
|---|---|
| $SO_2$ | $0.1 - 10.0$ ppmv |
| LWC | $0.1 - 2.5$ g m$^{-3}$ |
| TMI | $0.1 - 3.0$ µM |

**Table 6.** Contribution to sulphate production from different pathways of sulphur oxidation at varying initial concentration of $SO_2$

| $C_0, SO_2$ (ppmv) | Time (day) | $\Delta^{17}O(OH)$ (‰) | OH | $O_3$ | $H_2O_2$ | $O_2$/TMI | $\Delta^{17}O(S(VI), \text{dep.})_f$ (‰) |
|---|---|---|---|---|---|---|---|
| 0.01 | 1 | 0 | 39 | 0 | 55 | 6 | 0.38 |
| | 7 | 0 | 35 | 0 | 60 | 5 | 0.47 |
| 0.03 | 1 | 0 | 60 | 0 | 30 | 10 | 0.18 |
| | 7 | 0 | 54 | 0 | 40 | 6 | 0.32 |
| 0.1 | 1 | 0.3 | 63 | 0 | 22 | 15 | 0.15 |
| | 7 | 0.05 | 66 | 0 | 28 | 6 | 0.32 |
| 0.3 | 1 | 0.8 | 56 | 0 | 19 | 25 | 0.17 |
| | 7 | 0.3 | 69 | 0 | 22 | 9 | 0.29 |
| 1 | 1 | 2.6 | 43 | 0 | 15 | 42 | 0.20 |
| | 7 | 2 | 62 | 0 | 19 | 19 | 0.50 |
| 3 | 1 | 6.2 | 30 | 0 | 10 | 60 | 0.23 |
| | 7 | 5.6 | 51 | 0 | 14 | 35 | 0.77 |
| 10 | 1 | 11.5 | 17 | 0 | 5 | 77 | 0.19 |
| | 7 | 11.2 | 33 | 0 | 9 | 58 | 0.82 |

*Author contributions.* TG, SB, EM, and JS designed the study. SA provided the initial version of the model. TG carried out the research, and performed data analysis. TG, and SB wrote the manuscript with contributions from all authors. All authors have given approval to the final version of the manuscript.

*Competing interests.* The authors declare that they have no conflict of interest.

5 *Acknowledgements.* We would like to thank the two reviewers for their constructive comments. Discussions with Elena Maters and Gholamali Hoshiyaripour were also useful to address issues related to TMI concentrations in aqueous solution of in-plume water phases. The Agence Nationale de la Recherche (ANR) via contract 14-CE33-0009-02-FOFAMIFS is acknowledged for its financial support.

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
