# Peer review of "Photochemical box-modelling of volcanic $SO_2$ oxidation: isotopic constraints"

_Atmospheric Chemistry and Physics, 2018_

## Referee Comment (RC1) · Anonymous Referee #1 · 14 May 2018

Bekki et al. report calculations of the oxygen isotopic composition of sulfate formed in volcanic plumes with the goal of understanding why observations show very little O-17 enrichment. They use a box model including gas phase oxidation by OH and aqueous-phase oxidation by H2O2, O3 and TMI-catalysis. They conclude that oxidation by OH and TMI are the dominant pathways for secondary sulfate formation in volcanic plumes, and that the TMI pathway is necessary to explain the observations because OH has an O-17 enrichment in volcanic plumes. H2O2 is not important because it is quickly depleted by high concentrations of SO2, and O3 is not importance because of the acidic environment.

The only significant comment I have is that I think the paper should put more into a discussion of the implications of the lack of reactive halogen chemistry. Even in a volcanic eruption with zero halogens (is this possible?), entrainment of background air would supply some halogens. Could this be important? Perhaps it wouldn't take much fro background to have halogen recycling in this acidic environment. What are the implications for sulfate O-MIF formed in volcanic plumes?

Other than that, this manuscript is well written and I have only relatively minor comments that are detailed below.

One general comment: I think it would be good to helpful to highlight the observations of sulfate O-MIF, including the types of volcanos measured, in the paper. I suggest adding a Table describing the observations with appropriate references. As currently written it's hard for readers to compare model results with observations.

Abstract: I think a comment about reactive halogens belongs in the abstract. Perhaps say you are only considering volcanoes that are thought to have little halogen production.

Page 2 line 22: How much is "most"? The current thinking is that globally, DMS oxidation is the largest natural source of SO2 to the troposphere.

Page 3 line 3: The year for the Harris et al. reference is missing.

Page 5 line 1: How common are these types of volcanoes that are halogen-poor? Are the observations of sulfate O-17 excess from these types of volcanoes?

Page 6 equation 6: I think it would be helpful to explicitly show the equation for Ji. At least provide the page number where this can be found in the textbook that is cited.

Page 7 line 2: There is a missing end parentheses.

Page 7 R1: There is no other ion that can influence acidity? Ion balance equations are not the best way to calculate pH. In the future, perhaps it would be better to explicitly calculate the droplet pH in the model. This would certainly be necessary if these calculations were for ambient air, and not an SO2 rich volcanic plume.

Page 7: Why consider sulfate loss via deposition but not SO2? Globally, dry deposition of SO2 represents about half of tropospheric SO2 loss.

Page 12 line 28 and elsewhere: I think "gr" should be written as "g"

Page 13 line 6 and elsewhere: There are lots of citations for this textbook. It's better to cite original references when possible. This seems like an example of where you should be able to do this.

Page 16 line 27: I don't understand this sentence: "The sharp drop in H2O2…" I thought H2O2 concentrations decrease because of the high amounts of SO2.

Page 17 line 6: Does the O-17 excess of OH decrease as SO2 decreases? I think you should state why this is happening.

Page 17 lines 8-9: This sentence does not make sense. If you want to produce an O-MIF in sulfate, you need an oxidant with an O-MIF, not "null O-MIF".

Page 17 line 11: Perhaps remind the reader here that your calculations do not include halogens.

Page 20 line 8: "budget", not "budged"

In the figures, I don't understand why SO2 concentrations do not decrease over time. Sulfate increases, so mass balance suggests that SO2 should decreases, unless there is a continuous supply of SO2.

Figure 7: What is the y-axis unit?

Table 2: Needs a unit (%) for the 4th column.

---

## Referee Comment (RC2) · Anonymous Referee #2 · 13 Jul 2018

Galeazzo et al. have made developments to the atmospheric chemistry box model CiTTyCAT to investigate sulfate O-isotopes in the oxidation of volcanic SO2 emissions. Volcanic sulfur emissions are converted into sulfate by gas- and aqueous-phase pathways in the troposphere. Through a series of model sensitivity studies, Galeazzo et al. show that the oxidation of volcanic SO2 into sulfate cannot simply be assumed to follow the same dominant pathways as oxidation of SO2 in the background atmosphere. They demonstrate the importance of the TMI/O2 pathway in the model, that is consistent with sulfate isotope data. The study is novel in introducing a new modelling tool that enables a much more quantitative interpretation of volcanic sulfate isotopes in terms of the underlying atmospheric SO2 oxidation processes. It is a solid work that makes a substantial contribution to our understanding of the volcanic plume sulfur

chemistry. I find it very suitable for publication in ACP. Comments are given below.

Main comments:

1) Description of Isotopes data:

My main comment is to improve the description of the sulfate isotopes data used in the model comparison. The model development to include isotopes is well described, as are the sensitivity tests whose results are compared to reported sulfate isotope data leading to conclusions on oxidation pathways. The use of isotopes to study atmospheric pathways is also well described. However, fewer details are given about the existing/reported isotopes samples themselves or about variability in the isotopic data used for the model comparison. Even though these are existing reported data it would be useful to mention some more information, so the reader has a better idea of the data the model is being compared to. e.g. are these measured in-situ the plume or measured by sampling sulfate preserved in ash-deposits? From what kinds of eruptions to the troposphere, how many different volcanic emissions have been sampled, how far back in the past? A short description or visualization of the reported isotopes data and their variability given early in the manuscript (e.g. when introduced in page 4) would then enable the reader to better place the model results in context (e.g. when discussed later page 15 line 1-3, page 15 line 28, page 17 line 17).

2) Modelling Detail:

Page 12 line 16: Season and time of the simulations is given but details of the Pressure/Altitude, Temperature and Humidity for the model simulations should also be stated. These can affect the photolysis and reaction rates. For example: the reaction SO2 + OH + M is slower at lower pressures, higher altitudes. Species abundances are reported as mixing ratios (e.g. SO2 = 1 ppmv) in the model set-up but the concentration in the model hence reaction rates will also depend on atmospheric density, as well as temperature. Would the findings of the study be similar or very different for model simulations assuming a different injection altitude in the troposphere?

3) Always cloudy conditions?

Page 12 Line 27: You state that due to the large amounts of water injected during eruptions relatively high LWC can be expected i.e. cloudy conditions. This is true, but the abundance of volcanic H2O relative to background humidity will depend on how dilute the plume is. The relative abundances can be calculated for the chosen starting mixing ratio for SO2 (1 ppmv standard case, varying from 0.1 to 10 ppmv in sensitivity simulations) and assuming an example composition of a volcano plume as can be found in the literature (e.g. 80-90% H2O and up to a few % SO2). Would high LWC be expected for all of the SO2 dilution scenarios tested – or only if background conditions are also already at high RH/cloudy?

4) Conclusions - limitations:

A major result of this study is how important the TMI-O2 pathway is for oxidation of SO2 in volcanic plumes. The authors rightly emphasize on page 20 line 18 that uncertainty in Fe/Mn concentrations/speciation in volcanic plumes affects the oxidation by O2/TMI pathway – volcanic plume SO2 oxidation could be faster by this pathway if Fe/Mn concentrations are elevated.

Page20 Line 20: The absence of halogens is mentioned as a limitation of the model study. Whilst halogen emissions will affect oxidant concentrations, the authors are correct in pointing out that some volcanic emissions are halogen poor. Indeed, it is worth highlighting that two of the most important recent volcanic eruptions that impacted the troposphere on regional scales were actually halogen poor (relative to sulfur): the 2014-2015 Bardarbunga/Holuhraun eruption in Iceland and volcanic emissions from Kilauea, Hawaii such as the ongoing eruption.

One possible additional limitation of te study is that upon dispersion and dilution into a low RH atmosphere the volcanic plume may become more of an aerosol plume than a high LWC cloud as mentioned above. Under these conditions the particle phase would likely become a very acidic (sulfate-rich) volcanic aerosol instead of cloud droplets.

Could this affect the results and how? I would imagine that the highly acid conditions in the sulfate/sulphuric acid aerosol could act to limit extent of SO2 dissolution hence aqueous-phase pathways for oxidation of SO2. But the aerosols might also become more concentrated in Fe/Mn ions (particularly if the volcano emits significant Fe/Mn or ash) and thereby promote aqueous-phase oxidation of SO2 by TMI/O2.

5) Previous studies of SO2 oxidation:

Finally, it would be useful to discuss the findings from this isotopic modelling study in the context of some existing (non-isotopic) studies. For example, how do the simulations of SO2 oxidation compare to the SO2 oxidation rate observed by Kroll et al. Env. Sci. Tech. 2015 at Kilauea, Hawaii (in plume with quite high SO2 concentrations)? Also, it seems that this study's findings show some similarities but some differences to a recent review paper: Pattantyus et al. Review of sulfur dioxide to sulfate aerosol chemistry at KÄńlauea Volcano, Hawai'i, Atmospheric Environment, 2018. This could be because Pattantyus et al. considers a more dilute plume and assumes background atmospheric concentrations whereas Galeazzo et al. evaluates in more detail the SO2 oxidation chemistry in the concentrated plume by actively modelling the feedbacks on oxidant concentrations.

Some minor comments/language suggestions:

Page 2 line 13: "Once injected into the troposphere, volcanic SO2 is converted in a few days typically to H2SO4 by a range of gas-phase and liquid-phase reactions taking place in volcanic plumes and clouds (Chin and Jacob, 1996; Stevenson et al., 2003a)." This sentence appears to contradict the rest of the paper. The lifetime of a few days for SO2 is probably true for background atmospheric conditions but the lifetime of SO2 in volcanic plumes can be significantly longer due to the depletion of oxidants. Indeed that is one of the conclusions reached by the authors in the modelling of oxidation pathways according to the figures where SO2 persists over the week-long simulations.

Page 5. Line 7. Perhaps say our standard volcanic plume conditions (as not standard

background)

Page 5 line 14. Are species other than sulfate deposited in the model? If so, specify.

Page 6 Line 29, Page 7 line 16: Check labelling of Table 1/2.

Page 7 Line 1: "rate of SO2 dissolution". If I understood correctly should this rather be "extent of SO2 dissolution" is controlled by pH (as it is determined by equilibrium constants rather than rate constants)?

Page 9 Line 24: Perhaps clarify in the text that after reaction with ozone the remaining O and 2H are provided by reaction with water, - this is not explicitly clear in Table 2.

Page 10. The text of this section starts with the assumption that you already know the origins of OH. This might not be the case for all interdisciplinary readers. It would be better to explain at the beginning of this section the origins of OH eg. That OH is formed in the troposphere from photolysis of Ozone followed by reaction with H2O.

Page 10 Eq(19): Define k*OH+H2O in the text.

Page 13 Line 18: Do you mean: passive/quiescent degassing conditions?

Page 13 line 24: "It is widely recognised that SO2 is the compound emitted by volcanic activity that causes the widest climate impacts" Perhaps add: through its conversion into sulfate aerosols (as it is not SO2 itself that causes climate impacts)

Page 18 Line 26: I think it would be clearer to say -> mostly lie close to zero within measurements uncertainties. Also: Page 19 Line 22

---

## Author Comment (AC1) · 29 Sep 2018

We would like to thank the two anonymous reviewers for their helpful comments on this work.

**1  Reviewer suggestions and comments:**

**1.1  1st Comment**

The only significant comment I have is that I think the paper should put more into a discussion of the implications of the lack of reactive halogen chemistry. Even in a

volcanic eruption with zero halogens (is this possible?), entrainment of background air would supply some halogens. Could this be important? Perhaps it wouldn't take much from background to have halogen recycling in this acidic environment. What are the implications for sulfate O-MIF formed in volcanic plumes?

**Reply**

Halogens composition of volcanic emissions vary widely between different volcanic systems. Hotspot (such as Kīlauea) and rifting-plate volcanoes (such as Erta'ale in Ethiopia and Icelandic volcanoes) are characterised by low halogens contents (i.e. compared to the extent of sulphur emissions) whereas arc-volcanoes in subduction areas are characterised by emissions with relatively high halogens content (Aiuppa, 2009; Oppenheimer, 2013). Among volcanic eruptions with low halogens loading (compared to sulphur), one can cite the Bárðarbunga eruption in 2012-2014 (Ilyinskaya, 2017; Stefánsson, 2017) or the Kīlauea eruption in 2008 (Mather, 2012). Volcanic emissions from Kīlauea were characterised by $HCl/SO_2$ ratios (i.e. HCl most abundant halogen species) of the order of $10^{-2}$ (HCl concentrations 10-50 ppbv). Keep in mind that only volcanic plumes with liquid water (cloudy conditions) are considered. It is important for the chemistry of halogens.

Liquid water is assumed here to be present on ash surfaces or as water droplets. The question is about the fate of emitted halogens in water-rich plumes. Most of the halogens are emitted in the form of HCl and HBr which are very soluble species. In our model, pH values range between 1 and 2. These pH values are not low enough to limit significantly halogens dissolution in water (i.e. limited effect on halogens effective Henry's law coefficient). HCl and HBr should have relatively short lifetimes with respect to dissolution in cloudy conditions. The only way these soluble species can impact very significantly the plume chemistry is via rapid conversion into radical species, a process called halogen activation.

Halogens activation in the gas phase should be slow and would be inhibited by

HCl/HBr fast dissolution and deposition processes when liquid water is present (Roberts, 2009; von Glasow, 2010). The other pathway for halogen activation is heterogeneous. Experimental measurements show that halogens reactive uptakes on water are typically much slower than on sulphate aerosols (about 10 to 100 times slower) (Sander, 2006; Davidovits, 2009; Ammann, 2013). For example:

| Reaction | $\gamma_r$(sulpate) | $\gamma_r$(water) |
|---|---|---|
| $BrONO_2 + H_2O \rightarrow HOBr_{(aq.)} + HNO_{3(aq.)}$ | 0.8 | $3 \cdot 10^{-2}$ |
| $ClONO_2 + H_2O \rightarrow HOCl_{(aq.)} + HNO_{3(aq.)}$ | n.a | $2.5 \cdot 10^{-2}$ |

The second column reports reactive uptakes on sulphate aerosols $\gamma_r$(sulphate), while the third column reports reactive uptakes on water $\gamma_r$(water).

A similar trend can be assumed for other halogens reactions when reactive uptakes on water and sulphuric acid are compared. Halogen activation on water is expected to be much slower on a per molecule basis compared to activation on sulphate aerosols. Therefore, for our cloudy conditions, halogen activation should be relatively slow. Nonetheless, if very large amounts of halogens were emitted, the small fraction of halogen activation could be important for the chemistry. However, we are only considering low halogen emissions. Therefore, we have simply assumed that most of the volcanic HCl and HBr would be washed out from plumes. We accept that neglecting halogens might be an oversimplification for many volcanic plumes, especially for halogen-rich plumes. Only a full study about the role of volcanic halogens for halogen-rich plumes could answer this question. We are now more cautious in the text and stressing in the conclusions that the potential role of halogens should be explored.

Proposed change: "[. . .]**The focus here is on volcanic clouds that are rich in sulphur but poor in halogens, such in the case of intra-plate and rifting plate volcanoes (e.g. Nyarogongo in Congo, Erta'ale in Ethiopia, Kīlauea in Hawai'i) (Aiuppa et al., 2009; Oppenheimer et al., 2013).Volcanic eruptions with remarkable low**

halogens to sulphur emissions are the Holuhraun (Bárðarbunga) eruption of 2012-2014 in Iceland (Ilyinskaya et al., 2017; Stefánsson et al., 2017), and the Kīlauea eruption of 2008 in Hawai'i (Mather et al., 2012). In particular, HCl/SO$_2$ ratios of the order of 10$^{-2}$ have been observed for the Kīlauea eruption of 2008 (i.e. HCl $\approx$ 10-50 ppbv). [. . .]"

At the end of the conclusion section: "[. . .]**Overall, the present simulations might only be representative of degassing or eruptions with extremely low halogen emissions, typically originating from intraplate and rift volcanic activity. It is certainly worth exploring the potential impact of halogens in the case of halogen-rich eruptions, notably for volcanic plumes where water does not condense and hence only sulphate aerosols are present. Since the heterogeneous conversion of halogen halides into radicals is known to be fast on sulphate aerosols (references), halogens might impact significantly the plume chemistry and the isotopic composition of secondary sulphate under those conditions.** [. . .]"

**1.2   2nd Comment**

One general comment: I think it would be good to helpful to highlight the observations of sulfate O-MIF, including the types of volcanos measured, in the paper. I suggest adding a Table describing the observations with appropriate references. As currently written it's hard for readers to compare model results with observations.

**Reply**
A new table has now been added to enable to compare our results with oxygen isotopic measurements on tropospheric volcanic sulphate **(see attached figure)**.

The text has also been amended:

"[. . .] **Tropospheric volcanic sulphates of the present era distinguish themselves from other tropospheric sulphates by having a $\triangle^{17}O$ often close to zero (within the measurement error of about 0.1). This feature is found all over the world in sulphates collected from volcanic ashes of small and medium-size tropospheric explosive eruptions, independently from location, or geology of ash-deposits (Bao et al., 2003; Bindeman et al., 2007; Martin et al., 2014; see Table: 1). This is also the case for volcanic sulphate extracted from ash-deposits which are found very far from volcanoes, where secondary sulphate is expected to dominate.** [. . .]"

A second change is introduced in page 15 (lines 27-30), in order to recall the origins of experimental measurements. In this case, the values observed in sulphate aerosols collected at volcanic vents are also reported in the new table (Mather et al., 2006):

"[. . .] **The resulting model $\triangle^{17}O(S(VI))$ (i.e. from standard and sensitivity simulations) are compared to sulphate O-MIF found in tropospheric volcanic sulphates extracted from ash-deposits of small and medium-size tropospheric explosive eruptions of the present geological era (Bindeman et al., 2007; Martin et al., 2014; Bao, 2015), or in sulphate aerosols collected at volcanic vents, most certainly primary sulphate (Mather et al., 2006).** [. . .]"

**1.3 3rd Comment**

Page 2 line 22: How much is "most"? The current thinking is that globally, DMS oxidation is the largest natural source of $SO_2$ to the troposphere.

**Reply**

DMS is the largest natural source of $SO_2$, which however is a by-product of DMS oxidation (Chin, 1996,200). The largest direct source of natural $SO_2$ is volcanic activity, which roughly releases 10.4-13 Tg/year of $SO_2$ (Andres and Kasgnoc, 1998) mostly via quiescent degassing. We have rephrased this part:

Proposed change:

"[...]**Nowadays, anthropogenic $SO_2$ emissions outweigh those from natural sources (Smith et al., 2011). Volcanic emissions release about 10-13 $Tg \cdot y^{-1}$ of $SO_2$ to the atmosphere (Bates et al., 1992; Graf et al., 1998, Andres and Kasgnoc, 1998) and contribute to up to 10% to total sulphur emissions to the atmosphere (Stevenson et al., 2003a).** [...]"

1.4   4th Comment

Page 5 line 1: How common are these types of volcanoes that are halogen-poor? Are the observations of sulfate O-17 excess from these types of volcanoes?

**Reply**
Rift and hotspot volcanoes are usually characterised by halogen-poor emissions, because of the absence of subduction fluids in melts. In addition, volcanic emissions composition can widely change for a same volcano in relation to its erupting phases. Even for arc-volcanoes, it is possible to observe emissions with low halogen-to-sulphur ratios, notably for emissions from fumaroles (Aiuppa, 2009; Oppenheimer, 2013). Overall, we expect halogens-poor plumes to originate from both hotspot and intraplate eruptions. Unfortunately, most oxygen isotopic measurements are from arc-volcanoes (Bindeman et al., 2007; Martin et al., 2014; Martin, E. 2018). Some measurements are also from rifting volcanoes, notably from Icelandic eruptions (e.g. Gjálp eruption 1996).

Proposed change: Some examples of volcanic systems and recent volcanic eruptions

with low halogens/sulphur emissions are already provided in the first comment. (See 1st comment)

**1.5  5th comment**

Page 7 R1: There is no other ion that can influence acidity? Ion balance equations are not the best way to calculate pH. In the future, perhaps it would be better to explicitly calculate the droplet pH in the model. This would certainly be necessary if these calculations were for ambient air, and not an $SO_2$ rich volcanic plume.

**Reply**
We agree. In our case, sulphur is in excess and drives the pH of water. The effect of other ions is overwhelmed by the presence of very high S(VI) concentrations in water.

**1.6  6th comment**

Page 7: Why consider sulfate loss via deposition but not $SO_2$? Globally, dry deposition of $SO_2$ represents about half of tropospheric $SO_2$ loss.

**Reply**
There is a misunderstanding. The text was not clear about this point. We are only considering the fate of $SO_2$ in the core of a volcanic plume. Dry deposition as such is not expected to be important in the plume compared to wet deposition. Therefore, only wet deposition is accounted for in the model. Under those conditions, the wet deposition rate is determined by the scavenging of soluble species (including $SO_2$) by liquid water phases (followed either or not by oxidation). The model includes deposition of S(IV) and S(VI) species dissolved (including dissolved $SO_2$). More details are now

provided to clarify this point.

Proposed change:

"[. . .] **where $k_j$ the rate constant of the aqueous reaction between oxidant $C_j$ and relevant [S(IV)] species (see the list of aqueous oxidation reaction in Table: 2), and $k_d$ is the deposition coefficient of dissolved sulphur species. Dry deposition as such is not expected to be important in the plume itself compared to wet deposition for our cloudy conditions. Since only wet deposition is considered, only species dissolved in water phases such as aqueous S(IV) (SO$_{2(aq)}$ + HSO$_3^-$ + SO$_3^{2-}$) and S(VI) (HSO$_4^-$ + SO$_4^{2-}$) species are deposited in the model. The deposition is treated as a first order loss with k$_d$ = 2·$10^{-6}$ s$^{-1}$, equivalent to a characteristic time scale of 5.7 days (Stevenson et al., 2003a)** [. . .]"

**1.7  7th comment**

Page 17 line 6: Does the O-17 excess of OH decrease as SO$_2$ decreases? I think you should state why this is happening.

**Reply**

Proposed change:

"[. . .]  The final O-MIF in deposited sulphate is 1.1 permil, originating mostly from OH oxidation. **Recall that, when OH is generated via its main production channel, it carries an isotopic anomaly. Under common (non-volcanic) conditions, the OH anomaly is so rapidly erased by isotopic exchange with H$_2$O that, when OH reacts, it carries no anomaly. However, when SO$_2$ levels are very high, OH might react very quickly with SO$_2$ without having lost its anomaly by isotopic exchange.**

[Figure]

In this situation, the value of $\Delta^{17}O(OH)$ is determined by the competition between the $SO_2 + OH$ reaction and the OH isotopic exchange with $H_2O$. As $SO_2$ concentration decays with time, $\Delta^{17}O(OH)$ decreases because the $SO_2 + OH$ reaction slows down and becomes less competitive with respect to the isotopic exchange [. . .]"

**1.8 8th comment**

In the figures, I don't understand why $SO_2$ concentrations do not decrease over time. Sulfate increases, so mass balance suggests that $SO_2$ should decreases, unless there is a continuous supply of $SO_2$.

**Reply**
During S1 and S2 simulations $SO_2$ concentrations drop from an initial value of 1.5 ppmv, to roughly 1.2 ppmv with 7 days from injection; in S3 $SO_2$ concentration drops from 1.5 to 1 ppmv. Unfortunately, the small drop in concentration is not very noticeable with a logarithmic scale covering from pptv to ppmv concentrations. Therefore, for each figure showing the concentration evolution of key trace species, we have now reported in the figure captions the initial and final value of $SO_2$ concentration.

**1.9 References:**

Aiuppa, A. (2009). Degassing of halogens from basaltic volcanism: Insights from volcanic gas observations. Chem. Geol., 263(1-4):99–109.

Ammann, M., Cox, R. A., Crowley, J. N., Jenkin, M. E., Mellouki, A., Rossi, M. J., Troe, J., and Wallington, T. J. (2013). Evaluated kinetic and photochemical data for atmospheric chemistry: Volume VI heterogeneous reactions with liquid substrates. Atmos.

Chem. Phys., 13(16):8045–8228.

Andres, R. J. and Kasgnoc, a. D. (1998). A time-averaged inventory of subaerial volcanic sulfur emissions. J. Geophys. Res., 103(D19):25251.

Bao, H. (2015). Sulfate: A time capsule for Earth' s O2 , O3, and H2O. Chem. Geol., 395:108–118.

Bao, H., Thiemens, M. H., Loope, D. B., and Yuan, X. L. (2003). Sulfate oxygen-17 anomaly in an Oligocene ash bed in mid-North America: Was it the dry fogs? Geophys. Res. Lett., 30(16).

Bindeman, I. N., Eiler, J. M., Wing, B. A., and Farquhar, J. (2007). Rare sulfur and triple oxygen isotope geochemistry of volcanogenic sulfate aerosols. Geochim. Cosmochim. Acta, 71(9):2326–2343.

Chin, M., Jacob, D. J., Gardner, G. M., Foreman-Fowler, M. S., Spiro, P. A., and Savoie, D. L. (1996). A global three-dimensional model of tropospheric sulfate. J. Geophys. Res. Atmos., 101(D13):18667–18690.

Chin, M., Rood, R. B., Lin, S.-J., Müller, J.-F., and Thompson, A. M. (2000). Atmospheric sulfur cycle simulated in the global model GOCART: Model description and global properties. J. Geophys. Res. Atmos., 105(D20):24671– 24687.

Davidovits, P., Kolb, C. E., Williams, L. R., Jayne, J. T., and Worsnop, D. R. (2006). Mass accommodation and chemical reactions at gas-liquid interfaces. Chem. Rev., 106(4):1323–1354.

[revised manuscript text omitted]

---

## Author Comment (AC2) · 29 Sep 2018

We would like to thank the two anonymous reviewers for their helpful comments on this work.

**1 Answers to 2nd reviewer suggestions and comments:**

**1.1 Description of Isotopes data:**

My main comment is to improve the description of the sulfate isotopes data used in the model comparison. The model development to include isotopes is well described,

as are the sensitivity tests whose results are compared to reported sulfate isotope data leading to conclusions on oxidation pathways. The use of isotopes to study atmospheric pathways is also well described. However, fewer details are given about the existing/reported isotope samples themselves or about variability in the isotopic data used for the mode comparison. Even though these are existing reported data it would be useful to mention some more information, so the reader has a better idea of the data the model is being compared to. e.g. are these measured in-situ the plume or measured by sampling sulfate preserved in ash-deposits? From what kinds of eruptions to the troposphere, how many different volcanic emissions have been sampled, how far back in the past? A short description or visualization of the reported isotopes data and their variability given early in the manuscript (e.g. when introduced in page 4) would then enable the reader to better place the model results in context (e.g. when discussed later page 15 line 1-3, page 15 line 28, page 17 line 17).

**Reply**

We fully agree. A new table has now been added to enable to compare our results with oxygen isotopic measurements on tropospheric volcanic sulphate **(see attached figure)**.

The text has also been amended. The first change in page 4 (line 22) specifies the origins of most of volcanic sulphates from isotopic experimental measurements:

"[. . .] **Tropospheric volcanic sulphates of the present era distinguish themselves from other tropospheric sulphates by having a $\Delta^{17}O$ often close to zero (within the measurement error of about 0.1). This feature is found all over the world in sulphates collected from volcanic ashes of small and medium-size tropospheric explosive eruptions, independently from location, or geology of ash-deposits (Bao et al., 2003; Bindeman et al., 2007; Martin et al., 2014; see Table: 1). This is also the case for volcanic sulphate extracted from ash-deposits which are found very far from volcanoes, where secondary sulphate is expected to dominate.**

[. . .]"

A second change is introduced in page 15 (lines 27-30), in order to recall the origins of experimental measurements. In this case, the values observed in sulphate aerosols collected at volcanic vents are also reported in the new table (Mather, 2006):

"[. . .] **The resulting model $\Delta^{17}$O(S(VI)) (i.e. from standard and sensitivity simulations) are compared to sulphate O-MIF found in tropospheric volcanic sulphates extracted from ash-deposits of small and medium-size tropospheric explosive eruptions of the present geological era (Bindeman et al., 2007; Martin et al., 2014; Bao, 2015), or in sulphate aerosols collected at volcanic vents, most certainly primary sulphate (Mather et al., 2006).** [. . .]"

**1.2 Modelling Detail:**

Page 12 line 16: Season and time of the simulations is given but details of the Pressure/Altitude, Temperature and Humidity for the model simulations should also be stated. These can affect the photolysis and reaction rates. For example: the reaction $SO_2$ + OH + M is slower at lower pressures, higher altitudes. Species abundances are reported as mixing ratios (e.g. $SO_2$ = 1 ppmv) in the model set-up but the concentration in the model hence reaction rates will also depend on atmospheric density, as well as temperature. Would the findings of the study be similar or very different for model simulations assuming a different injection altitude in the troposphere?

**Reply**
The pressure/altitude and temperatures of simulations are now given in Page 12.

"[. . .] Since most of volcanoes are situated in remote areas with their peaks close to the free troposphere, or, at least, with volcanic plumes often ending up in the free

troposphere, **the environmental conditions are chosen to be representative of the lower free troposphere with temperature set at 283.15 K, and pressure fixed at 640 mbar (about 3 km altitude). Since we consider cloudy conditions, the relative humidity is set to 100%.**

Furthermore, concentrations of reactive species are also set to typical levels found in the tropical lower free troposphere: $O_3$ = 45 ppbv and $H_2O_2$ = 0.1 ppbv. [. . .]"

It is difficult to claim that our conclusions are valid for the entire troposphere. For example, the competition on the OH budget between the isotopic exchange with $H_2O$ and the reaction with $SO_2$ (crucial for OH isotopic signature) would change according to the water vapour concentration and hence the altitude. If lower (higher) altitudes were considered, the water vapour concentration is expected to be much higher (lower) and therefore higher (lower) $SO_2$ levels would compete with the isotopic exchange with water. In the same way, $O_3$ levels would also vary depending on the altitude and region. The purpose of the paper is to identify and explore the key processes for sulphate production and isotopic composition in a volcanic plume. It is, indeed, a process study not meant to cover the full range of possible tropospheric conditions. Ideally, this kind of investigation should be done through a global 3-D model.

**1.3 Always cloudy conditions?**

Page 12 Line 27: You state that due to the large amounts of water injected during eruptions relatively high LWC can be expected i.e. cloudy conditions. This is true, but the abundance of volcanic $H_2O$ relative to background humidity will depend on how dilute the plume is. The relative abundances can be calculated for the chosen starting mixing ratio for $SO_2$ (1 ppmv standard case, varying from 0.1 to 10 ppmv in sensitivity simulations) and assuming an example composition of a volcano plume as can be found in the literature (e.g. 80-90% $H_2O$ and up to a few % $SO_2$). Would high LWC be expected for all of the $SO_2$ dilution scenarios tested – or only if background conditions

are also already at high RH/cloudy?

**Reply**
It is a good point. The conditions prevailing for some of the volcanic plumes might not be favourable for cloud formation, which would depend on the amount of water vapour injected, the local temperature and plume dilution. First, since we consider plumes within the lower free troposphere where temperatures are relatively low (about 15K below than surface temperatures), water condensation is more likely. Second, the mixing between the plume and background is assumed to be weak (timescale of 10 days). Therefore, the plume tends to remain relatively dense during the 7 days simulations, which favours maintaining cloudy conditions. Nonetheless, water clouds do not form in all the volcanic plumes, notably plumes from degassing. Therefore, we have added a comment in the conclusions to point out that our model results so far are only applicable to cloudy volcanic plumes:

"[. . .] We stress that oxidation by $O_2$/TMI is poorly constrained in model simulations because of the lack of measurements of TMI aqueous concentrations in volcanic plumes. **It is worth pointing out that our model results are only applicable to cloudy volcanic plumes. Nonetheless, water clouds do not always form in volcanic plumes, notably during passive degassing. It would be interesting to also consider cloud-free plumes where the condensed phase is concentrated sulphuric acid within sulphate aerosols. In particular, these particles have a chemical reactivity radically different from water droplets.** [. . .]"

1.4   Conclusions

One possible additional limitation of the study is that upon dispersion and dilution into a low RH atmosphere the volcanic plume may become more of an aerosol plume than a high LWC cloud as mentioned above. Under these conditions the particle phase would

]

likely become a very acidic (sulfate-rich) volcanic aerosol instead of cloud droplets. Could this affect the results and how? I would imagine that the highly acid conditions in the sulfate/sulphuric acid aerosol could act to limit extent of $SO_2$ dissolution hence aqueous-phase pathways for oxidation of $SO_2$. But the aerosols might also become more concentrated in Fe/Mn ions (particularly if the volcano emits significant Fe/Mn or ash) and thereby promote aqueous-phase oxidation of $SO_2$ by TMI/$O_2$.

**Reply**

We agree with this comment and the possible implications for the plume chemistry. At this stage, it is difficult to speculate in the paper without having carried out this type of simulations. As stated above, few lines have been now added in the conclusion to highlight this limitation and suggest exploring cloud-free plumes.

"[. . .] We stress that oxidation by $O_2$/TMI is poorly constrained in model simulations because of the lack of measurements of TMI aqueous concentrations in volcanic plumes. **It is worth pointing out that our model results are only applicable to cloudy volcanic plumes. Nonetheless, water clouds do not always form in volcanic plumes, notably during passive degassing. It would be interesting to also consider cloud-free plumes where the condensed phase is concentrated sulphuric acid within sulphate aerosols. In particular, these particles have a chemical reactivity radically different from water droplets.** [. . .]"

1.5   References:

Bao, H. (2015). Sulfate: A time capsule for Earth' s O2 , O3, and H2O. Chem. Geol., 395:108–118.

Bao, H., Thiemens, M. H., Loope, D. B., and Yuan, X. L. (2003). Sulfate oxygen-17 anomaly in an Oligocene ash bed in mid-North America: Was it the dry fogs? Geophys.

Res. Lett., 30(16).

Bindeman, I. N., Eiler, J. M., Wing, B. A., and Farquhar, J. (2007). Rare sulfur and triple oxygen isotope geochemistry of volcanogenic sulfate aerosols. Geochim. Cosmochim. Acta, 71(9):2326–2343.

Martin, E. (2018). Volcanic Plume Impact on the Atmosphere and Climate: O and S-Isotope Insight into Sulfate Aerosol Formation. 1991:1–23.

Martin, E., Bekki, S., Ninin, C., and Bindeman, I. (2014). Volcanic sulfate aerosol formation in the troposphere. J. Geophys. Res. Atmos., 119(22): 12,660–12,673.

Mather, T. A., McCabe, J. R., Rai, V. K., Thiemens, M. H., Pyle, D. M., Heaton, T. H. E., Sloane, H. J., and Fern, G. R. (2006). Oxygen and sulfur isotopic composition of volcanic sulfate aerosol at the point of emission. J. Geophys. Res. Atmos., 111(18):1–9.
* * *
[Figure]

| Volcano & Date of Eruption | Sample distance (km) | Source | $\Delta^{17}O$ (‰) | Reference |
|---|---|---|---|---|
| **Popocatépetl** (Mexico), 2008 | 25 | ash | 0.35 | (Martin et al., 2014) |
| **Spurr** (Alaska, USA), 1992 | 265 | ash | -0.14 | (Martin et al., 2014) |
| **Fuego** (Guatemala), 1974 | 57 | ash | -0.04 | (Martin et al., 2014) |
| **Negro Cerro** (Nicaragua), 1947 | 12 | ash | -0.06 | (Martin et al., 2014) |
| **Parícutin** (Mexico), 1948 | 5 | ash | 0.13 | (Martin et al., 2014) |
| **Mt. St. Helens** (USA), 1980 | 400 | ash | 0.02 | (Martin et al., 2014) |
| **Gjálp** (Iceland), 1998 | < 30 | ash | -0.07 | (Martin et al., 2014) |
| **Pinatubo** (Philippines), 1991 | < 50 | ash | -0.04 | (Bindeman et al., 2007) |
| **Pinatubo** (Philippines), 1991 | < 50 | ash | 0.19 | (Bindeman et al., 2007) |
| **Spurr** (USA), 1953 | n.a. | ash | 0.06 | (Bindeman et al., 2007) |
| **Vesuvius** (Italy), 1872 | n.a. | ash | -0.07 | (Bao et al., 2003) |
| **Popocatépetl** (Mexico), 1997 | n.a. | ash | -0.08 | (Bao et al., 2003) |
| **Spurr** (USA), 1992 | n.a. | ash | 0.06 | (Bao et al., 2003) |
| **Fuego** (Guatemala), 1974 | 55 | ash | -0.03 | (Bao et al., 2003) |
| **Pinatubo** (Philippines), 1991 | n.a. | anhydrite from pumice | -0.01 | (Bao et al., 2003) |
| **Santorini** (Greece), Minoan age | n.a. | pumice + ash | 0.09 | (Bao et al., 2003) |
| **Masaya** (Nicaragua), 2003 | 0 | aerosols | 0.1 | (Mather et al., 2006) |
| **Masaya** (Nicaragua), 2003 | 0 | aerosols | 0.2 | (Mather et al., 2006) |

\* Refer to (Martin, 2018) for a more extensive description regarding oxygen isotopic anomalies measured in tropospheric volcanic sulphate of present and past geological eras.

**Fig. 1.** Oxygen isotopic composition of volcanic sulphates from different tropospheric eruptions of the present geological era.